# Enriching Time Series Representation: Integrating a Noise-Resilient Sampling Strategy with an Efficient Encoder Architecture

## Abstract

Time series analysis has been an important research area for decades, and with the advent of foundation models, it has witnessed an explosive surge in interest. Contrastive self-supervised learning stands out as a powerful technique to learn representations capable of solving a wide range of downstream tasks. However, there have been several challenges that persist. First, there is no previous work explicitly considering noise, which is one of the critical factors affecting the efficacy of time series tasks. Second, there is a lack of efficient yet lightweight encoder architectures that can learn informative representations robust to various downstream tasks. To fill in these gaps, we initiate a novel sampling strategy that promotes consistent representation learning with the presence of noise in natural time series. In addition, we propose an encoder architecture that utilizes dilated convolution within the Inception block to create a scalable and robust network architecture with a wide receptive field. Experiments demonstrate that our method consistently outperforms state-of-the-art methods in forecasting, classification, and abnormality detection tasks, e.g. ranks first over two-thirds of the classification UCR datasets, with only $40\%$ of the parameters compared to the second-best approach.

## 1 Introduction

**Motivation.** Time series data, found in finance, medicine, and engineering, require essential analysis in practical applications Shumway et al. (2000); Small (2005); Peng et al. (1995); Box et al. (2015). Labeling such data is often difficult and costly due to their intricate, uninterpretable patterns, especially in privacy-sensitive fields like healthcare and finance Kayaalp (2018); Mucherino et al. (2009); Ta et al. (2016). Unsupervised learning offers a solution, enabling the learning of informative representations for diverse downstream tasks without labels. While unsupervised representation learning thrives in computer vision and natural language processing Caron et al. (2018); Zhang et al. (2016); Pathak et al. (2016); Mikolov et al. (2013); Joulin et al. (2016); Wang et al. (2018), its potential in time series remains underexplored. Recognizing this gap in existing works Franceschi et al. (2019); Tonekaboni et al. (2021); Eldele et al. (2021); Yang & linda Qiao (2022); Yue et al. (2022), we address the time series representation learning challenge.

Our framework design follows two crucial principles: (1) efficiency (ensuring accurate downstream task performance by capturing essential time series characteristics) and (2) scalability (being lightweight to handle practical, lengthy, high-dimensional, and high-frequency time series data).

**Related literature and our approach.** Prior research on representation learning in time series data has predominantly focused on employing the self-supervised contrastive learning technique Franceschi et al. (2019); Tonekaboni et al. (2021); Eldele et al. (2021); Yang & linda Qiao (2022); Yue et al. (2022), which consists of two main components: *sampling strategy* and *encoder architecture*.

Existing sampling strategies revolve around time series' *invariance characteristics*, encompassing temporal invariance Kiyasseh et al. (2021); Tonekaboni et al. (2021); Franceschi et al. (2019), transformation and augmentation invariance Tang et al. (2020); Yang & linda Qiao (2022); Zhang et al. (2022), and contextual invariance Eldele et al. (2021); Yue et al. (2022). For instance, TNC Tonekaboni et al. (2021) leverages temporal invariance for positive pair sampling but faces limitations

in real-time applicability due to quadratic complexity. BTSF Yang & linda Qiao (2022) combines dropout and spectral representations, yet its efficiency relies on dropout rate and time instance length. Some studies Eldele et al. (2021); Yue et al. (2022) maintain contextual invariance, with Yue et al. (2022) focusing on consistent representations across different contexts (i.e., time segments). However, this may risk losing surrounding context information due to temporal masking. A common shortcoming that emerges in existing methods is the way they handle noise during the learning of time series representations, which significantly impacts task accuracy (Song et al. (2022); Wen et al. (2019)). Many of these methods either disregard the noisy nature of time series or implicitly rely on the ability of Neural Networks to deal with this undesired signal, instead of explicitly addressing them upon learning time series representations. Acknowledging this, we devise a sampling strategy guided by the principle that the presence of noise in the series should not hinder the functionality of our framework. Ideally, it should generate consistent representations whether provided with noise-free or raw series, highlighting *noise-resiliency characteristics*. To achieve this, we employ a spectrum-based low-pass filter to generate correlated yet distinct views of each input time series. The corresponding representations are then guided by our proposed system of loss functions. These loss functions effectively align embeddings of the raw-augmented couplets to attain desired noise invariance, while simultaneously preserving important information through a Triplet-based regularization term. The advantages of this combination are twofold: (1) the filter preserves key characteristics such as trend and seasonality, ensuring deterministic and interpretable representations, while eliminating noise-prone high-frequency components; (2) the loss system stably directs the network in improving noise resilience and retaining information, leading to a significant enhancement in downstream task performance.

In addition to effective sampling strategies, the importance of robust encoder architectures for generating versatile time series representations is often underestimated by researchers. Common methods include linear models Zeng et al. (2023a), auto-encoders Choi et al. (2016), sequence-to-sequence models Gupta et al. (2018); Lyu et al. (2018), and Convolution-based designs like Causal Convolution Bai et al. (2018b); Wan et al. (2019); Franceschi et al. (2019) and Dilated Convolution Franceschi et al. (2019); Yue et al. (2022). Yet, these approaches may struggle with long-term dependencies, particularly for extensive time series data. Alternatively, Transformer-based models and their variations Kitaev et al. (2020); Li et al. (2019b); Zhou et al. (2021); Fan et al. (2023); Cao et al. (2023); Nie et al. (2022) are adopted to address long-term dependencies, but can be computationally demanding and vulnerable to collapse on specific tasks or data Dong et al. (2021); Shwartz-Ziv & Armon (2022); Sun et al. (2017); Zeng et al. (2023b). In response, we propose an efficient and scalable encoder framework, combining the strengths of Dilated Convolution and Inception idea. While Dilated Convolution achieves a broad receptive field without excessive depth, the Inception concept, which utilize multi-scale filters, effectively automate the process of choosing dilation factors, captures sequential correlations across scales. This dual approach balances representation effectiveness and model scalability. In addition, we enhance the vanilla Inception framework by introducing a novel convolution-based aggregator and extra skip connections within the Inception block, boosting its ability to capture long-term dependencies in input time series.

**Our contributions.** In this study, we introduce *CoInception*, a noise-resilient, robust, and scalable representation learning framework for time series. Our main contributions are as follows.

- We are the first to directly address the adverse effects of noise in learning time series representations. Specifically, we introduce an effective combination of noise-resilient sampling strategy and loss system that enables learning consistent representations even in the presence of noise in natural time series data.

- We present a robust and scalable encoder that leverages the advantages of well-established Inception blocks and the Dilation concept in convolution layers. With this, we can maintain a lightweight, shallow, yet robust framework while ensuring a wide receptive field of the final output.

- We conducted comprehensive experiments to evaluate the efficacy of CoInception and analyze its behavior. Our empirical results demonstrate that our approach outperforms the current state-of-the-art methods on three major time series tasks: forecasting, classification, and anomaly detection.

## 2 PROPOSED METHOD

In this section, we majorly focus on the CoInception framework. We first present mathematical definitions of different time series problems in Sec. 2.1. Following, the technical details and training methodology of our method would be discussed in Sec. 2.2 and 2.3.

### 2.1 PROBLEM FORMULATION

The majority of natural time series can be represented as a continuous or discrete stream. Without loss of generality, we only consider the discrete series (continuous ones can be discretized through a quantization process). Let $\mathcal{X} = \{\mathbf{x}_1, \mathbf{x}_2, \ldots, \mathbf{x}_n\}$ be such a dataset with $n$ sequences, where $\mathbf{x}_i \in \mathbb{R}^{M \times N}$ ($M$ is sequence length and $N$ is number of features), our goal is to obtain the corresponding latent representations $\mathcal{Z} = \{\mathbf{z}_1, \mathbf{z}_2, \ldots, \mathbf{z}_n\}$, in which $\mathbf{z}_i \in \mathbb{R}^{M \times H}$ ($M$ is sequence length and $H$ is desired latent dimension). The time resolution of the learnt representations is kept intact as the original sequences, which has been shown to be more beneficial for adapting the representations to many downstream tasks Yue et al. (2022). Our ultimate goal of learning the latent representations is to adapt them as the new input for popular time series tasks, where we define the objectives below. Let $\mathbf{z}_i = \begin{bmatrix} z_i^1, \ldots, z_i^M \end{bmatrix}$ be the learned representation for each segment,

- **Forecasting** requires the prediction of corresponding $T$-step ahead future observations $\mathbf{y}_i = \begin{bmatrix} y_i^{M+1}, \ldots, y_i^{M+T} \end{bmatrix}$;
- **Classification** aims at identifying the correct label in the form $\mathbf{p}_i = [p_1, \ldots, p_C]$, where $C$ is the number of classes;
- **Anomaly detection** determines whether the last time step $x_i^M$ (corresponding to $z_i^M$) is an abnormal point (streaming evaluation protocol - Ren et al. (2019)).

From now on, without further mention, we would implicitly exclude the index number $i$ for readability.

### 2.2 COINCEPTION FRAMEWORK

Adopting an unsupervised contrastive learning strategy, CoInception framework can be decomposed into three distinct components: (1) Sampling strategy, (2) Encoder architecture, and (3) Loss function. Figure 1 illustrates the overall architecture.

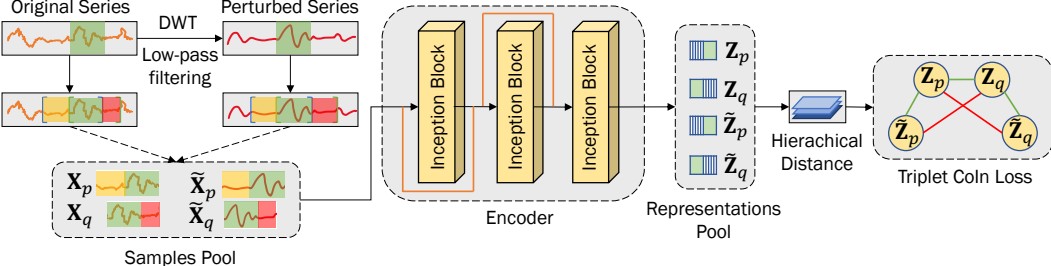

Figure 1: **Overview of the proposed CoInception framework**. We use samples from *Representation Pool* in different downstream tasks.

SAMPLING STRATEGY

Natural time series often contain noise, represented as a random process that oscillates independently alongside the main signal (e.g., white noise Parzen (1966); Pagano (1974)). For illustration, series $\mathbf{x}$ could be decomposed into disentangled components $\hat{\mathbf{x}}$ and $\mathbf{n}$, depicting the original signal and an independent noise component, respectively. While existing approaches treat the signal and noise separately when only the noise factor varies (e.g., $\mathbf{n} \rightarrow \tilde{\mathbf{n}}$), we argue that the high-frequency noise-like elements, which appear in high frequency spectrum of the original series, provide little to

no meaningful information, and may degrade the accuracy of the downstream tasks severely. This realization is well aligned with studies Huang et al. (2024); Wang et al. (2022); Woo et al. (2022) emphasizing the utilization of disentangled components of raw series, such as seasonality or trends, which exhibit persistence in a long-term context. In addition, to validate the sensitivity of existing frameworks with noise, we conduct a toy experiment with a synthesized series (upper left plot in Fig. 2) and its disturbed version with two noise signals added (upper right plot of Fig. 2). We adopt cosine similarity as the correlation measurement. Considering the high correlation (0.961) between noisy and noiseless series, together with their negligible visual differences (Fig. 2), we expect the fundamental characteristics of learnt representations to remain intact. However, an existing state-of-the-art (SOTA) framework Yue et al. (2022) fails to exhibit such a strong relation (correlation reduced to 0.837, visually demonstrated by two bottom trajectories), highlighting its noise susceptibility. In contrast, CoInception's outcomes (two middle trajectories) show strong consistence (correlation of 0.983), capturing the original sine wave's harmonic shift even in noisy scenario. This underscores the importance of *noise resiliency* in representations, resisting such high-frequency signals. Figure

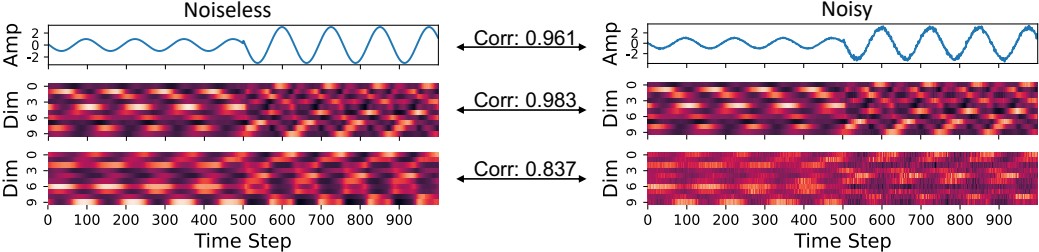

Figure 2: **Output representations for toy time series containing high-frequency noise.** CoInception's results can still capture the periodic characteristics regardless of the presence of noise.

1 depicts an overview of the proposed sampling strategy. We leverage Discrete Wavelet Transform (DWT) as a parameter-free low-pass filter Skodras (2015) to generate a perturbed version $\tilde{\mathbf{x}}$ of original series $\mathbf{x}$. The DWT filter convolves the input series with a set of shifted wavelet functions to generate coefficients representing their contributions at various intervals, before downsampling the result by the factor of 2. This filter is applied $L$ times, corresponding to $L$ levels of decomposition, with the output of previous iteration be the input of the next one. This process essentially segregates the original series into $L + 1$ distinct frequency bands, in which the low-frequency approximation coefficients reflect the overall trend of the data, whereas the high-frequency detail coefficients represent noise-like components. Here, $L = \lfloor \log_2(\frac{M}{K}) \rfloor$ is the maximum useful level of the decomposition, where $K$ is the length of chosen mother wavelet. Mathematically, let $\mathbf{g}$ and $\mathbf{h}$ denote the low-pass and its quadrature-mirror high-pass filters, respectively. The working of DWT filter at the $j$-th level and position $n$ is as follows.

$$\mathbf{x}^j[n] = (\mathbf{x}^{j-1} * \mathbf{g}^j)[n] = \sum_k \mathbf{x}^{j-1}[k]\mathbf{g}^j[2n-k],$$
$$\mathbf{d}^j[n] = (\mathbf{x}^{j-1} * \mathbf{h}^j)[n] = \sum_k \mathbf{x}^{j-1}[k]\mathbf{h}^j[2n-k].$$

where $*$ represents convolution operator, $k$ denotes the shifted coefficient, $\mathbf{g}^j$ and $\mathbf{h}^j$ are the low-pass and high-pass filter coefficients, $\mathbf{x}^j$ and $\mathbf{d}^j$ represent the approximation and detail coefficients. Following, to create a perturbed version of the input series, we intentionally retain only the significant values in the detail coefficients $\mathbf{d}^j$ ($j \in \{1, \ldots, L\}$), while masking out unnecessary (potentially noise) values, which result in perturbed detail coefficients $\tilde{\mathbf{d}}^j$ as follows.

$$\tilde{\mathbf{d}}^j = \left[ \frac{d_k}{|d_k|} \times \max(|d_k| - \gamma, 0) \,\middle|\, k \in \{1, \ldots, \text{len}(\mathbf{d}^j)\} \right]. \tag{1}$$

With this strategy, we define a cutting threshold $\gamma$ to be proportional to the maximum value of input series $\mathbf{x}$ by a hyper-parameter $\alpha < 1$, i.e., $\gamma = \alpha \times \max(\mathbf{x})$. Subsequently, the reconstruction process involves the approximation coefficient $\mathbf{x}^{\mathbf{L}}$ and set of perturbed detail coefficients $\left\{ \tilde{\mathbf{d}}^1, \ldots, \tilde{\mathbf{d}}^L \right\}$ using the inverse Discrete Wavelet Transform (iDWT), producing the modified series $\tilde{\mathbf{x}}$. Details of

this process are provided in Appendix A.1. The sampling phase concludes with the implementation of random cropping on both $\mathbf{x}$ and $\tilde{\mathbf{x}}$, resulting in overlapping segments $\langle \mathbf{x}_p; \mathbf{x}_q \rangle$ and $\langle \tilde{\mathbf{x}}_p; \tilde{\mathbf{x}}_q \rangle$. These segments are subsequently utilized by the CoInception encoder (Section 2.2).

INCEPTION-BASED DILATED CONVOLUTION ENCODER

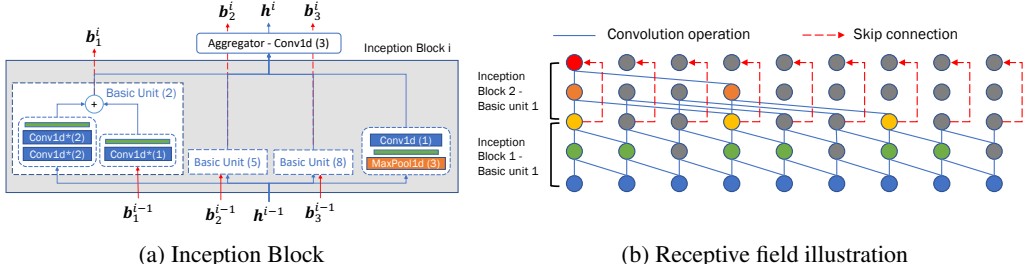

| (a) Inception Block | (b) Receptive field illustration |

Figure 3: Illustration of the Inception block and the accumulated receptive field upon stacking.

In pursuit of an architecture that strikes a balance between robustness and efficiency, we deliver the CoInception encoder which integrates principles from Dilated Convolution and the Inception concept. Previous studies Bai et al. (2018a); Franceschi et al. (2019); Yue et al. (2022) highlight the robustness of stacked Dilated Convolutional Networks in various tasks, emphasizing their potential. The strength of this architecture lies in its ability to retain low scale of networks parameters, while maintaining robustness via a large accumulative receptive field. However, a key weakness arises in the selection of dilation factors, posing a trade-off between effectiveness and efficiency. Small factors reduce the parameter-efficient gain, while large factors risk focusing too much on broad contextual information, neglecting local details. To address this, our Inception-based design automates the incorporation of different dilation factors into a single layer, called Inception block (Fig.3a). Specifically, within each block, there are several Basic units encompasing 1D convolutional layers of varying filter lengths and dilation factors. This configuration enables the encoder to consider input segments at diverse scales and resolutions.

In addition, apart from existing Inception-based models Ismail Fawaz et al. (2020), Wu et al. (2022), we introduce two additional modifications to improve robustness and scalability, without sacrificing design simplicity: (1) an aggregator layer, and (2) extra skip connections (i.e., red arrows in Fig. 3b). Regarding the aggregator, beyond the aim of reducing the number of parameters as in Szegedy et al. (2016); Ismail Fawaz et al. (2020), it is intentionally placed after the Basic units to better combine the features $\mathbf{b}$ produced by those layers, producing aggregated representation $\mathbf{h}$. Moreover, with the stacking nature of Inception blocks in our design, the aggregator can still inherit the low-channel-dimension output of the previous block, just like the conventional Bottleneck layer. Furthermore, we introduce extra skip connections that interconnect the outputs of these units across different Inception blocks, denoted as modification (2). These skip connections have two-fold benefits of serving as shortcut links for stable gradient flow and gluing up the Basic units of different Inception blocks, making the entire encoder horizontally and vertically connected. In this way, our CoInception framework can be seen as a set of multiple Dilated Convolution experts, with much shallower depth and equivalent receptive fields compared with ordinary stacked Dilated Convolution networks Bai et al. (2018a); Franceschi et al. (2019); Yue et al. (2022). Mathematically, let $k_u$ be the base kernel size (the numbers in bracket of Fig.3a) for a Basic unit $u$ within the $i^{th}$ Inception block (1-indexed), the dilation factor and the receptive field are calculated as $d_u^i = (2k-1)^{i-1}; r_u^i = (2k-1)^i$. Illustrated in Figure 3b is the accumulative receptive field associated with the Basic unit featuring a base kernel size of 2, at the first and second Inception blocks.

## 2.3 HIERARCHICAL TRIPLET LOSS

Aligned with the sampling strategy outlined in Section 2.2, a system of loss functions are deployed to attain robust and noise-resilient representation. We integrate the concept of hierarchical loss Yue et al. (2022) and triplet loss Chechik et al. (2010); Hoffer & Ailon (2015) to enhance *noise resiliency*,

incorporating a variation of *contextual consistency* inspired by Yue et al. (2022) (details in Appendix A.1). For simplicity in annotation, we use $\langle \mathbf{z}_p; \mathbf{z}_q \rangle$ and $\langle \tilde{\mathbf{z}}_p; \tilde{\mathbf{z}}_q \rangle$ to denote the representations of the actual overlapping segments between the sampled couplets $\langle \mathbf{x}_p; \mathbf{x}_q \rangle$ and $\langle \tilde{\mathbf{x}}_p; \tilde{\mathbf{x}}_q \rangle$ (green timestamps in Fig.1). With this, the *noise-resilient* characteristic is ensured by minimizing the distances between representations of the original segments and their perturbed views - $\langle \mathbf{z}_p; \tilde{\mathbf{z}}_p \rangle$ and $\langle \mathbf{z}_q; \tilde{\mathbf{z}}_q \rangle$. In parallel, the embeddings $\mathbf{z}_p$ and $\mathbf{z}_q$ should also be close in latent space to preserve the *contextual consistency*. To model the distance within a couplet, we incorporate both instance-wise loss Franceschi et al. (2019) ($\mathcal{L}_{ins}$) and temporal loss Tonekaboni et al. (2021) ($\mathcal{L}_{temp}$). The combination of these two forms the consistency loss $\mathcal{L}_{con}$.

$$\mathcal{L}_{temp}(\mathbf{z}_p,\mathbf{z}_q) = \frac{-1}{BT} \sum_{b,t}^{B,T} \log \frac{\exp(z_p^{b,t} \cdot z_q^{b,t})}{\sum_{\tilde{t}}^{T} \left( \exp(z_p^{b,t} \cdot z_q^{b,\tilde{t}}) + \mathbb{1}_{t \neq \tilde{t}} \exp(z_p^{b,t} \cdot z_p^{b,\tilde{t}}) \right)},$$

$$\mathcal{L}_{ins}(\mathbf{z}_p,\mathbf{z}_q) = \frac{-1}{BT} \sum_{b,t}^{B,T} \log \frac{\exp(z_p^{b,t} \cdot z_q^{b,t})}{\sum_{\tilde{b}}^{B} \left( \exp(z_p^{b,t} \cdot z_q^{\tilde{b},t}) + \mathbb{1}_{b \neq \tilde{b}} \exp(z_p^{b,t} \cdot z_p^{\tilde{b},t}) \right)},$$

$$\mathcal{L}_{con}(\mathbf{z}_p,\mathbf{z}_q) = \mathcal{L}_{temp}(\mathbf{z}_p,\mathbf{z}_q) + \mathcal{L}_{ins}(\mathbf{z}_p,\mathbf{z}_q).$$

In addition, to further enhance the reliability of learned representations, we propose to enforce an auxiliary criteria, based on the following observation. Apart from the previously mentioned pairs, extra couplets can be formed by comparing the representations of an original segment with a perturbed version in a different context, e.g., $\langle \mathbf{z}_p; \tilde{\mathbf{z}}_q \rangle$. $\mathbf{z}_p$ and $\mathbf{z}_q$ are from the same original samples, forming the common region of two segments $\mathbf{x}_p$ and $\mathbf{x}_q$. Conversely, $\tilde{\mathbf{z}}_p$ or $\tilde{\mathbf{z}}_q$ arises from the overlap of two perturbed segments $\tilde{\mathbf{x}}_p$ and $\tilde{\mathbf{x}}_q$. Therefore, it is reasonable to expect the proximity between $\mathbf{z}_p$ and the unaltered representation $\mathbf{z}_q$ is greater than that of $\mathbf{z}_p$ and the modified counterpart $\tilde{\mathbf{z}}_q$. This condition could also help in mitigating the over-smoothing effect potentially caused by the DWT-low pass filter; details about this strength are mentioned in Appendix A.3. We incorporate this observation as a constraint in the final loss function (denoted as $\mathcal{L}_{triplet}$) in the format of a triplet loss as follows.

$$\mathcal{L}_{triplet}(l_{pq}, l_{p\tilde{q}}, l_{\tilde{p}q}, \epsilon, \zeta) = \epsilon \times \frac{l_{pq} + l_{p\tilde{p}} + l_{q\tilde{q}}}{3} + (1-\epsilon) \times \max(0, 2 \times l_{pq} - l_{p\tilde{q}} - l_{\tilde{p}q} + 2 \times \zeta), \quad (2)$$

where $l_{pq}$ represents $\mathcal{L}_{con}(\mathbf{z}_p, \mathbf{z}_q)$, $l_{p\tilde{q}}$ is $\mathcal{L}_{con}(\mathbf{z}_p, \tilde{\mathbf{z}}_q)$ and similar notations for remaining terms. $\epsilon < 1$ is the balance factor for two loss terms, while $\zeta$ denotes the triplet margin. To ensure the CoInception framework can handle inputs of multiple granularity levels, we adopt a hierarchical strategy similar to Yue et al. (2022) with our $\mathcal{L}_{triplet}$ loss. We describe the details in Appendix A.3 - Algorithm 1.

## 3 EXPERIMENTS

In this section, we empirically validate the effectiveness of the CoInception framework and compare the results with the recent state of the arts. We consider three major tasks, including forecasting, classification, and anomaly detection, as in Section 2.1. Our evaluation encompasses multiple benchmarks, some of which also follow unsupervised training strategy and target multiple tasks: (1) **TS2Vec** Yue et al. (2022) learns to preserve contextual invariance across multiple time resolutions using a sampling strategy and hierarchical loss; (2) **TS-TCC** Eldele et al. (2021) combines cross-view prediction and contrastive learning tasks by creating two views of the raw time series data using weak and strong augmentations; (3) **TNC** Tonekaboni et al. (2021) tailors for time series data that forms positive and negative pairs from nearby and distant segments, respectively, leveraging the stationary properties of time series. For better comparison, in all of our experiments, we highlight best results in **bold** and **red**, and second best results are in blue.

### 3.1 TIME-SERIES FORECASTING

**Datasets & Settings.** For this experiment, the same settings as Zhou et al. (2021) are adopted for both short-term and long-term forecasting. In addition to the representative works, CoInception is further compared with studies that delicately target the forecasting task, such as Informer Zhou et al. (2021), StemGNN Cao et al. (2020), LogTrans Li et al. (2019a), N-BEATS Oreshkin et al. (2019), and LSTnet Lai et al. (2018). Among these frameworks, Informer Zhou et al. (2021) is a supervised model which requires no extra regressor to process its produced representations. For other unsupervised benchmarks, a linear regression model is trained using the $L2$ norm penalty, with the

learned representation $\mathbf{z}$ as input to directly predict future values. To ensure a fair comparison with works that only generate instance-level representations, only the $M^{th}$ timestep representation $z^M$ produced by the CoInception framework is used for the input segment. The evaluation of the forecast result is performed using two metrics, namely Mean Square Error (MSE) and Mean Absolute Error (MAE). For the datasets used, the Electricity Transformer Temperature (ETT) Zhou et al. (2021) datasets are adopted together with the UCI Electricity Trindade (2015) dataset.

Table 1: Multivariate time series forecasting results on MSE.

| T | TS2Vec | TS-TCC | TNC | Informer | StemGNN | LogTrans | CoInception | T | TS2Vec | TS-TCC | TNC | Informer | StemGNN | LogTrans | CoInception |
|---|---|---|---|---|---|---|---|---|---|---|---|---|---|---|---|
| *ETTh1:* | | | | | | | | *ETTm1:* | | | | | | | |
| 24 | 0.599 | 0.653 | 0.632 | 0.577 | 0.614 | 0.686 | **0.461** | 24 | 0.443 | 0.473 | 0.429 | **0.323** | 0.620 | 0.419 | 0.384 |
| 48 | 0.629 | 0.720 | 0.705 | 0.685 | 0.748 | 0.766 | **0.512** | 48 | 0.582 | 0.671 | 0.623 | **0.494** | 0.744 | 0.507 | 0.552 |
| 168 | 0.755 | 1.129 | 1.097 | 0.931 | **0.663** | 1.002 | 0.683 | 96 | 0.622 | 0.803 | 0.749 | 0.678 | 0.709 | 0.768 | **0.561** |
| 336 | 0.907 | 1.492 | 1.454 | 1.128 | 0.927 | 1.362 | **0.829** | 288 | 0.709 | 1.958 | 1.791 | 1.056 | 0.843 | 1.462 | **0.623** |
| 720 | 1.048 | 1.603 | 1.604 | 1.215 | - | 1.397 | **1.018** | 672 | 0.786 | 1.838 | 1.822 | 1.192 | - | 1.669 | **0.717** |
| *ETTh2:* | | | | | | | | *Electricity:* | | | | | | | |
| 24 | 0.398 | 0.883 | 0.830 | 0.720 | 1.292 | 0.828 | **0.335** | 24 | 0.287 | 0.278 | 0.305 | 0.312 | 0.439 | 0.297 | **0.234** |
| 48 | 0.580 | 1.701 | 1.689 | 1.457 | 1.099 | 1.806 | **0.550** | 48 | 0.307 | 0.313 | 0.317 | 0.392 | 0.413 | 0.316 | **0.265** |
| 168 | 1.901 | 3.956 | 3.792 | 3.489 | 2.282 | 4.070 | **1.812** | 168 | 0.332 | 0.338 | 0.358 | 0.515 | 0.506 | 0.426 | **0.282** |
| 336 | 2.304 | 3.992 | 3.516 | 2.723 | 3.086 | 3.875 | **2.151** | 336 | 0.349 | 0.357 | 0.349 | 0.759 | 0.647 | 0.365 | **0.301** |
| 720 | **2.650** | 4.732 | 4.501 | 3.467 | - | 3.913 | 2.962 | 720 | 0.375 | 0.382 | 0.447 | 0.969 | - | 0.344 | **0.331** |

**Results.** Due to limited space, we only present the multivariate forecasting results on MSE in Table 1, while the full results for both the univariate and multivariate scenario can be found in the Appendix C.1. Apparently, the proposed CoInception framework achieves the best results in most scenarios over all 4 datasets in the multivariate setting. The numbers indicate that our method outperforms existing state-of-the-art methods in most cases. Furthermore, the Inception-based encoder design results in a CoInception model with only $40\%$ number of parameters compared with the second-best approach (see Table 2).

## 3.2 TIME-SERIES CLASSIFICATION

**Datasets & Settings.** For the classification task, we follow the settings in Franceschi et al. (2019) and train an RBF SVM classifier on instance-level representations generated by our baselines. However, since CoInception produces timestamp-level representations for each data instance, we utilize the strategy from Yue et al. (2022) to ensure a fair comparison. Specifically, we apply a global MaxPooling operation over $\mathbf{z}$ to extract the instance-level vector representing the input segment. We assess the performance of all models using two metrics: prediction accuracy and the area under the precision-recall curve (AUPRC). We test the proposed approach against multiple benchmarks on two widely used repositories: the UCR Repository Dau et al. (2019) with 128 univariate datasets and the UEA Repository Bagnall et al. (2018) with 30 multivariate datasets. To further strengthen our empirical evidence, we additionally implement a K-nearest neighbor classifier equipped with DTW Chen et al. (2013) metric, along with T-Loss Franceschi et al. (2019) and TST Zerveas et al. (2021) beside the aforementioned SOTA approaches.

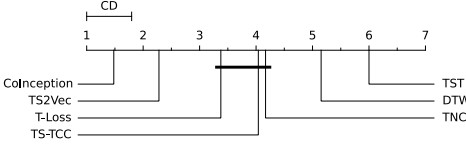

Figure 4: Critical Difference Diagram comparing different classifiers on 125 Datasets from UCR Repository with the confidence level of 95%.

Table 2: Time series classification results.

| Dataset | UCR repository | | | UEA repository | | |
|---|---|---|---|---|---|---|
| | Accuracy | Rank | Parameter | Accuracy | Rank | Parameter |
| DTW | 0.72 | 5.15 | - | 0.65 | 3.86 | - |
| TNC | 0.76 | 4.17 | - | 0.68 | 4.58 | - |
| TST | 0.64 | 6.00 | 2.88M | 0.64 | 5.27 | 2.88M |
| TS-TCC | 0.76 | 4.04 | 1.44M | 0.68 | 3.86 | 1.44M |
| T-Loss | 0.81 | 3.38 | 247K | 0.67 | 3.75 | 247M |
| TS2Vec | 0.83 | 2.28 | 641K | 0.71 | 2.96 | 641K |
| **CoInception** | **0.84** | **1.48** | **206K** | **0.72** | **1.89** | **206K** |

**Results.** Evaluation results of our proposed CoInception framework on UCR and UEA repositories are presented in Table 2. It is important to highlight that the results presented here pertain exclusively to 125 datasets within the UCR repository and 29 datasets in the UEA repository. The remainings are omitted to ensure fair comparisons among different baselines. We provide additional details in the Appendix C.2. For 125 univariate datasets in the UCR repository, CoInception ranks first in a majority

of 86 datasets, and for 29 UEA datasets it produces the best classification accuracy in 16 datasets. In this table, we also add a detailed number of parameters for every framework when setting a fixed latent dimension of 320. With the fusion of dilated convolution and Inception strategy, CoInception achieves the best performance while being much more lightweight (2.35 times) than the second best framework Yue et al. (2022). We also visualize the critical difference diagram Demšar (2006) for the Nemenyi tests on 125 UCR datasets in Figure 4. Intuitively, in this diagram, classifiers connected by a bold line indicate a statistically insignificant difference in average ranks. As suggested, CoInception makes a clear improvement gap compared with other SOTAs in average ranks.

## 3.3 TIME-SERIES ANOMALY DETECTION

**Datasets & Settings.** For this task, we adopt the protocols introduced by Ren et al. (2019); Yue et al. (2022). However, we make three forward passes during the evaluation process. In the first pass, we mask $x^M$ and generate the corresponding representation $z_1^M$. The second pass puts the input segment $\mathbf{x}$ through DWT low-pass filter (Section 2.2) to generate the perturbed segment $\tilde{\mathbf{x}}$, before getting the representation $z_2^M$. The normal input is used in the last pass, and $z_3^M$ is its corresponding output. Accordingly, we define the abnormal score as $\alpha_M = \frac{1}{2}\left(\|z_1^M - z_3^M\|_1 + \|z_2^M - z_3^M\|_1\right)$. We keep the remaining settings intact as Ren et al. (2019); Yue et al. (2022) for both normal and cold-start experiments. Precision (P), Recall (R), and F1 score (F1) are used to evaluate anomaly detection performance. We use the Yahoo dataset Laptev et al. (2015) and the KPI dataset Ren et al. (2019) from the AIOPS Challenge. Additionally, we compare CoInception with other SOTA unsupervised methods that are utilized for detecting anomalies, such as SPOT Siffer et al. (2017), DSPOT Siffer et al. (2017), DONUT Xu et al. (2018), and SR Ren et al. (2019) for normal detection tasks, as well as FFT Rasheed et al. (2009), Twitter-AD Vallis et al. (2014), and Luminol Luminol (2015) for cold-start detection tasks that require no training data.

Table 3: Time series abnormaly detection results.

| Dataset | Metrics | Normal Setting | | | | | | Cold-start Setting | | | | | |
|---------|---------|------|-------|-------|------|--------|------------|------|-----------|---------|------|--------|------------|
| | | SPOT | DSPOT | DONUT | SR | TS2Vec | **CoInception** | FFT | Twitter-AD | Luminol | SR | TS2Vec | **CoInception** |
| Yahoo | F1 | 0.338 | 0.316 | 0.026 | 0.563 | 0.745 | **0.769** | 0.291 | 0.245 | 0.388 | 0.529 | 0.726 | **0.745** |
| | Precision | 0.269 | 0.241 | 0.013 | 0.451 | 0.729 | 0.790 | 0.202 | 0.166 | 0.254 | 0.404 | 0.692 | 0.733 |
| | Recall | 0.454 | 0.458 | 0.825 | 0.747 | 0.762 | 0.748 | 0.517 | 0.462 | 0.818 | 0.765 | 0.763 | 0.754 |
| KPI | F1 | 0.217 | 0.521 | 0.347 | 0.622 | 0.677 | **0.681** | 0.538 | 0.330 | 0.417 | 0.666 | 0.676 | **0.682** |
| | Precision | 0.786 | 0.623 | 0.371 | 0.647 | 0.929 | 0.933 | 0.478 | 0.411 | 0.306 | 0.637 | 0.907 | 0.893 |
| | Recall | 0.126 | 0.447 | 0.326 | 0.598 | 0.533 | 0.536 | 0.615 | 0.276 | 0.650 | 0.697 | 0.540 | 0.552 |

**Results.** Table 3 presents a performance comparison of various methods on the Yahoo and KPI datasets using F1 score, precision, and recall metrics. We observe that CoInception outperforms existing SOTAs in the main F1 score for all two datasets in both the normal setting and the cold-start setting. In addition, CoInception also reveals its ability to perform transfer learning from one dataset to another, through steady enhancements in the empirical result for cold-start settings. This transferability characteristic is potentially a key to attaining a general framework for time series data. Futher experiments are presented in the Appendix C.7.

## 4 ANALYSIS

### 4.1 ABLATION ANALYSIS

In this experiment, we analyze the impact of different components on the overall performance of the CoInception framework. We designed three variations: (1) **Excluding noise-resilient sampling**, which follows the sampling strategy and hierarchical loss from Yue et al. (2022); (2) **Excluding Dilated Inception block**, where a stacked Dilated Convolution network is used instead of our CoInception encoder; and (3) **Excluding triplet loss**, which omits the triplet-based term from our $\mathcal{L}_{triplet}$ calculation. Our experiments encompass all three tasks, and the results are summarized in Table 4. For the classification task, we present average metrics across 5 UCR datasets and 5 UEA datasets, details about which is discussed in Appendix C.4. Regarding the forecasting task, we conducted univariate experiments on the ETTm1 dataset, with results averaged over various forecasting lengths, encompassing both short-term and long-term forecasting. In the anomaly

detection task, scores for the Yahoo dataset in normal setting are provided. Overall, substantial drops in performance are observed across all three versions in the primary time series tasks. The exclusion of noise-resilient sampling led to a performance decrease from $8\%$ in classification to $15\%$ in anomaly detection. The removal of the Dilated Inception-based encoder resulted in up to $9\%$ performance decline in anomaly detection, while the elimination of triplet loss contributed to performance reductions ranging from $4\%$ to $17\%$.

Table 4: Ablation analysis for the proposed CoInception framework.

|  | CoInception (1) | CoInception (2) | CoInception (3) | **CoInception** |
|---|---|---|---|---|
| *Classification:* | | | | |
| Acc. | 0.645 (- **8.51%**) | 0.661 (- **6.24%**) | 0.624 (- **11.48%**) | 0.705 |
| AUC. | 0.704 (- **9.04%**) | 0.726 (- **6.20%**) | 0.691 (- **10.72%**) | 0.774 |
| *Forecasting:* | | | | |
| MSE | 0.067 (- **8.95%**) | 0.065 (- **6.15%**) | 0.064 (- **4.68%**) | 0.061 |
| MAE | 0.178 (- **2.81%**) | 0.180 (- **3.88%**) | 0.177 (- **2.26%**) | 0.173 |
| *Anomaly Detection:* | | | | |
| F1 | 0.646 (- **15.99%**) | 0.704 (- **8.45%**) | 0.636 (-**17.29%**) | 0.769 |
| P. | 0.607 (- **23.16%**) | 0.720 (- **8.86%**) | 0.581 (-**26.45%**) | 0.790 |
| R. | 0.692 (- **7.48%**) | 0.689 (- **7.88%**) | 0.701 (- **6.28%**) | 0.748 |

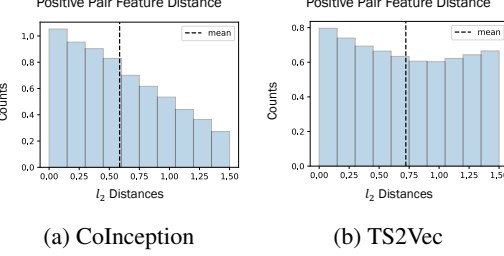

(a) CoInception    (b) TS2Vec

Table 5: **Alignment analysis.** Distribution of $l_2$ distance between features of positive pairs.

## 4.2 ALIGNMENT AND UNIFORMITY

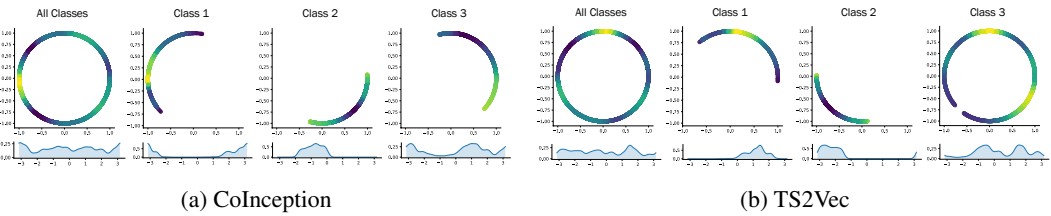

(a) CoInception    (b) TS2Vec

Figure 5: **Uniformity analysis.** Feature distributions with Gaussian kernel density estimation (KDE) (above) and von Mises-Fisher (vMF) KDE on angles (below).

We assess the learned representations via two qualities: Alignment and Uniformity, as proposed in Wang & Isola (2020). Alignment measures feature similarity across samples, ensuring insensitivity to noise in positive pairs. Uniformity aims to retain maximum information by minimizing the intra-similarities of positive pairs and maximizing inter-distances of negative pairs while maintaining a uniform feature distribution. Fig. 5 summarizes the alignment of testing set features for the StarLightCurves dataset generated by CoInception and TS2Vec Yue et al. (2022). Generally, CoInception's features exhibit a more closely clustered distribution for positive pairs. CoInception has smaller mean distances and decreasing bin heights as distance increases, unlike TS2Vec. In figure 5, we plot feature distributions using Gaussian kernel density estimation (KDE) and von Mises-Fisher (vMF) KDE for angles. CoInception demonstrates superior uniform characteristics for the entire test set representation, as well as better clustering between classes. Representations of different classes reside on different segments of the unit circle.

## 5 CONCLUSION

We introduce CoInception, a framework designed to tackle the challenges of robust and efficient time series representation learning. Our approach generates representations that are resilient to noise by utilizing a DWT low-pass filter. By incorporating Inception blocks and dilation concepts into our encoder framework, we strike a balance between robustness and efficiency. CoInception empirically outperforms state-of-the-art methods across a range of time series tasks including forecasting, classification and abnormality detection. About the limitation, we do recognize several hyper-parameters in our frameworks, which we believe extra efforts should be needed for particular tasks or datasets for achieving the best performance. For future work, we aim to explore the transferability of our approach and enhance its foundational characteristics for time series analysis.

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

# Enriching Time Series Representation: Integrating a Noise-Resilient Sampling Strategy with an Efficient Encoder Architecture
# Appendix, ICLR 2024

## A    CoInception Supplement Details

### A.1    Sampling Strategy

This section cover the working of invert DWT low-pass filter Skodras (2015) to reconstruct perturbed series $\tilde{\mathbf{x}}$. This process involves combining the approximation coefficients $\mathbf{x}^{\mathbf{L}}$ and perturbed detail coefficients $\left\{\tilde{\mathbf{d}}^1, \ldots, \tilde{\mathbf{d}}^L\right\}$ obtained during the decomposition process.

By utilizing the inverse low-pass and high-pass filters, the original signal can be reconstructed from these coefficients. Let $\tilde{\mathbf{g}}$ and $\tilde{\mathbf{h}}$ represent these low- and high-pass filters, respectively. The mathematical operations for the DWT reconstruction filters, which recover the perturbed signal $\tilde{\mathbf{x}}$ at level $j$ and position $n$, can be represented as follows:

$$\tilde{\mathbf{x}}^{j-1}[n] = (\hat{\mathbf{x}}^j * \tilde{\mathbf{g}}^j)[n] + (\tilde{\mathbf{d}}^j * \tilde{\mathbf{h}}^j)[n]$$
$$= \sum_k \hat{\mathbf{x}}^j[2n-k]\tilde{\mathbf{g}}^j[k] + \sum_k \tilde{\mathbf{d}}^j[2n-k]\tilde{\mathbf{h}}^j[k],$$

where

$$\begin{cases} \tilde{\mathbf{x}}^L & = \mathbf{x}^L \\ \hat{\mathbf{x}}^j & = \texttt{Upsampling}(\tilde{\mathbf{x}}^j, 2). \end{cases}$$

In these equations, $\hat{\mathbf{x}}^j$ represents the upsampled approximation at level $j$. The upsampling process $\texttt{Upsampling}$ involves inserting zeros between the consecutive coefficients to increase their length, effectively expanding the signal toward the original length of input series. Following, the upsampled coefficients are convolved with the corresponding reconstruction filters $\tilde{\mathbf{g}}^j$ and $\tilde{\mathbf{h}}^j$ to obtain the reconstructed signal $\tilde{\mathbf{x}}^{j-1}$ at the previous level.

This recursive filtering and upsampling process is repeated until the maximum useful level of decomposition, $L = \lfloor \log_2(\frac{M}{K}) \rfloor$, is reached. Here, $M$ represents the length of the original signal and $K$ is the length of the mother wavelet. By iteratively applying the reconstruction filters and combining the coefficients obtained after upsampling, the original signal can be reconstructed, gradually restoring both the overall trend (approximation) and the high-frequency details captured by the DWT decomposition.

To maintain the characteristic of *context invariance*, we employ a variation of the approach proposed with TS2Vec Yue et al. (2022). Specifically, we choose to rely solely on random cropping for generating overlapping segments, without incorporating temporal masking. This decision is based on recognizing several scenarios that could undermine the effectiveness of this strategy. Firstly, if heavy masking is applied, it may lead to a lack of explicit context information. The remaining context information, after extensive occlusion, might be insufficient or unrepresentative for recovering the masked timestamp, thus impeding the learning process. Secondly, when dealing with data containing occasional abnormal timestamps (e.g., level shifts), masking these timestamps in both overlapping segments (Figure 6) can also hinder the learning progress since the contextual information becomes non-representative for inference.

According to the findings discussed in Yue et al. (2022), random cropping is instrumental in producing position-agnostic representations, which helps prevent the occurrence of *representation collapse* when

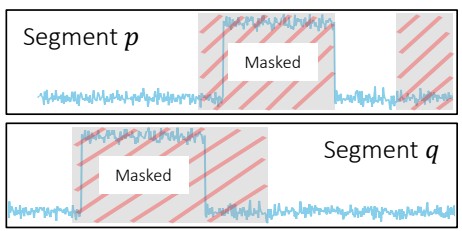

Figure 6: **Temporal Masking collapse**. An illustration for a case in which temporal masking would hinder training progress.

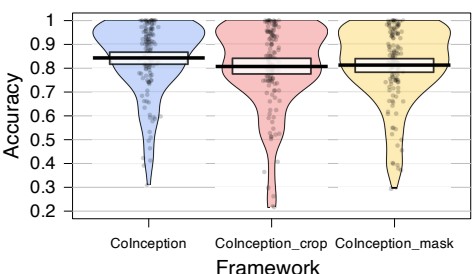

Figure 7: **Ablation in sampling strategy**. Accuracy distribution of different variants over 128 UCR dataset.

using temporal contrasting. This is attributed to the inherent capability of Convolutional networks to encode positional information in their learned representations Islam et al. (2020); Chen (2015), thereby mitigating the impact of temporal contrasting as a learning strategy. As the Inception block in CoInception primarily consists of Convolutional layers, the adoption of random cropping assumes utmost importance in enabling CoInception to generate meaningful representations.

The accuracy distribution for 128 UCR datasets across various CoInception variations is illustrated in Figure 7. These variations include: (1) the ablation of Random Cropping, where two similar segments are used instead, and (2) the inclusion of temporal masking on the latent representations, following the approach in Yue et al. (2022). As depicted in the figure, both variations exhibit a decrease in overall performance and a higher variance in accuracy across the 128 UCR datasets, compared with our proposed framework.

## A.2 INCEPTION-BASED DILATED CONVOLUTION ENCODER

While the main structure of our CoInception Encoder is a stack of Inception blocks, there are some additional details discussed in this section.

Before being fed into the first Inception block, the input segments are first projected into a different latent space, other than the original feature space. We intentionally perform the mapping with a simple Fully Connected layer.

$$\theta : \mathbb{R}^{M \times N} \to \mathbb{R}^{M \times K}$$
$$\theta(\mathbf{x}) = \mathbf{W}\mathbf{x} + \mathbf{b}$$

The benefits of this layer are twofold. First, upon dealing with high-dimensional series, this layer essentially act as a filter for dimensionality reduction. The latent space representation retains the most informative features of the input segment while discarding irrelevant or redundant information, reducing the computational burden on the subsequent Inception blocks. This layer make CoInception more versatile to different datasets, and ensure its scalability. Second, the projection by Fully Connected layer help CoInception enhance its transferability. Upon adapting the framework trained with one dataset to another, we only need to retrain the projection layer, while keeping the main stacked Inception layers intact.

To provide a clearer understanding of the architecture depicted in Inception block 3a, we will provide a detailed interpretation. In our implementation, each Inception block consists of three Basic units. Let the outputs of these units be denoted as $\mathbf{b}_1$, $\mathbf{b}_2$, and $\mathbf{b}_3$. To enhance comprehension, we will use the notation $\mathbf{b}_j$ to represent these three outputs collectively. Additionally, we will use $\mathbf{m}$ to denote the output of the Maxpooling unit, and $\mathbf{h}$ to represent the overall output of the entire Inception block. The following formulas outline the operations within the $i^{th}$ Inception block.

$$\begin{aligned}
\mathbf{b}_k^i &= \texttt{Conv1d}\star(\sigma(\texttt{Conv1d}\star(\mathbf{h}^{i-1}))) \\
&\quad + \sigma(\texttt{Conv1d}\star(\mathbf{b}_k^{i-1})), \\
\mathbf{m}^i &= \sigma(\texttt{Conv1d}\star(\mathbf{b}_k^{i-1})), \\
\mathbf{h}^i &= \texttt{Aggregator}(\texttt{Concat}(\mathbf{b}_k^i, \mathbf{m}^i)).
\end{aligned} \tag{3}$$

In these equations, $\sigma$ represent the *LeakyReLU* activation function, which is used throughout the CoInception architecture.

## A.3 HIERARCHICAL TRIPLET LOSS

**Mitigating over-smoothing effect.** In our pursuit to uphold the noise-resilience and contextual-invariance features of learned representations, we acknowledge a potential challenge in the learning process—the over-smoothing effect. This occurs when a low-pass filter eliminates excessive high-frequency information. Under these circumstances, the Triplet loss regularization term plays a crucial role in our CoInception framework. It automatically mitigates the need for strict alignment between the representations of raw and perturbed series by giving higher priority to fostering proximity between embeddings of overlapping segments.

**Hierarchical Triplet Loss** Algorithm 1 describe the procedure to calculate the hierarchical triplet loss mentioned in the manuscript. With this loss, we cover and maintain the objectives set out in various resolutions of input time series.

---

**Algorithm 1** Hierarchical Triplet Loss Calculation

**Input**:
$\triangleright$ $\mathbf{z}_i, \mathbf{z}_j$ - latent representations of $i^{th}$ and $j^{th}$ segments;
$\triangleright$ $\tilde{\mathbf{z}}_i, \tilde{\mathbf{z}}_j$ - latent representations of $i^{th}$ and $j^{th}$ **perturbed** segments;
$\triangleright$ $\epsilon$ - Balance factor between instance loss and temporal loss;
$\triangleright$ $\zeta$ - Triplet loss margin.
**Output**:
$\triangleright$ $l_{3hier}$ - Hierarchical triplet loss value

```
 1: function HIERTRIPLETLOSS()
 2:     Initialize l_{3hier} ← 0; r ← 0                        ▷ Running variable
 3:     while time_dimension(z_i) > 1 do
 4:                                                            ▷ Loop with reduced time resolution
 5:         l_{ij} ← L_con(z_i, z_j);
 7:         l_{iĩ} ← L_con(z_i, z_j);
 9:         l_{jj̃} ← L_con(z_j, z̃_j)
10:                                                            ▷ Losses for main couplets
12:         l_{ij̃} ← L_con(z_i, z̃_j);
14:         l_{ĩj} ← L_con(z̃_i, z_j)
15:                                                            ▷ Losses for supporting couplets
17:         l_{3hier} ← l_{3hier} + L_triplet(l_{ij}, l_{ij̃}, l_{ĩj}, ε, ζ);
18:
20:         z_i ← max_pool_1d(z_i);
22:         z_j ← max_pool_1d(z_j)
24:         z̃_i ← max_pool_1d(z̃_i);
26:         z̃_j ← max_pool_1d(z̃_j);
27:                                                            ▷ Reducing the time resolution
29:         d ← d + 1
30:     end while
31:     l_{3hier} ← l_{3hier}/d
32:     return l_{3hier}
33: end function
```

---

# B IMPLEMENTATION DETAILS

## B.1 ENVIRONMENT SETTINGS

All implementations and experiments are performed on a single machine with the following hardware configuration: an $64-$core Intel Xeon CPU with a GeForce RTX 3090 GPU to accelerate training.

Our codebase primarily relies on the *PyTorch 2.0* framework for deep learning tasks. Additionally, we utilize utilities from *Scikit-learn, Pandas,* and *Matplotlib* to support various functionalities in our experiments.

### B.2 CoInception's Reproduction

**Sampling Strategy.** In our current implementation for CoInception, we employ the Daubechies wavelet family Daubechies (1992), known for its widespread use and suitability for a broad range of signals Daubechies (1990); Vaidyanathan (2006); Mallat (1999). Specifically, we utilize the Daubechies D4 wavelets as both low and high-pass filters in CoInception across all experiments, as mentioned in Dolabdjian et al. (2002). It is important to note, however, that our selection of the mother wavelet serves as a reference, and it is advisable to invest additional effort in choosing the optimal wavelets for specific datasets Dolabdjian et al. (2002). Such careful consideration may further enhance the accuracy of CoInception for specific tasks.

**Inception-Based Dilated Convolution Encoder.** In our experiments, we incorporate three Inception blocks, each comprising three Basic units. The base kernel sizes employed in these blocks are 2, 5, and 8 respectively. For non-linear transformations, we utilize the $LeakyReLU$ activation function consistently across the architecture. To ensure fair comparisons across all benchmarks, we maintain a constant latent dimension of 64 and a final output representation size of 320.

**Hierarchical Triplet Loss.** In the calculation of $\mathcal{L}_{triplet}$ (Eq. 2), several hyperparameters are utilized. The balance factor $\epsilon$ is assigned a value of 0.7, indicating a higher weight distribution towards minimizing the distance between positive samples. The triplet term serves as an additional constraint and receives a relatively smaller weight. For the triplet term itself, the margin $\eta$ is set to 1.

### B.3 Baselines' Reproduction

Due to the extensive comparison of CoInception with numerous baselines, many of which are specifically designed for particular tasks, we have chosen to reproduce results for a selected subset while inheriting results from other relevant works. Specifically, we reproduce the results from three works that focus on various time series tasks, namely TS2Vec Yue et al. (2022), TS-TCC Eldele et al. (2021), and TNC Tonekaboni et al. (2021). The majority of the remaining results are directly sourced from Yue et al. (2022), Franceschi et al. (2019), Zhou et al. (2021), Ren et al. (2019), and Yang & linda Qiao (2022).

## C Further Experiment Results and Analysis

### C.1 Time Series Forecasting

**Additional details.** During the data processing stage, z-score normalization is applied to each feature in both the univariate and multivariate datasets. All reported results are based on scores obtained from these normalized datasets. In the univariate scenario, additional features are introduced alongside the main feature, following a similar approach as described in Zhou et al. (2021); Yue et al. (2022). These additional features include *minute*, *hour*, *day of week*, *day of month*, *day of year*, *month of year*, and *week of year*. For the train-test split, the first 12 months are used for training, followed by 4 months for validation, and the last 4 months for three ETT datasets, following the methodology outlined in Zhou et al. (2021). In the case of the Electricity dataset, a ratio of $60 - 20 - 20$ is used for the train, validation, and test sets, respectively, following Yue et al. (2022).

After the completion of the unsupervised training phase, the learned representations are evaluated using a forecasting task, following a protocol similar to Yue et al. (2022). A linear regression model with an $L_2$ regularization term $\alpha$ is employed. The value of $\alpha$ is chosen through a grid search over the search space $\{0.1, 0.2, 0.5, 1, 2, 5, 10, 20, 50, 100, 200, 500, 1000\}$.

**Additional results.** The full results for the univariate and multivariate forecasting experiments are presented in Table 6 and Table 7, respectively. For both circumstances, CoInception demonstrates its superiority in every testing dataset, in most configurations for the output number of forecasting timesteps (highlighted with **bold, red numbers**).

Table 6: Univariate time series forecasting results. Best results are bold and highlighted in **red**, and second best results are in blue.

| Dataset | T | TS2Vec | | TS-TCC | | TNC | | Informer | | N-BEATS | | LogTrans | | CoInception | |
|---|---|---|---|---|---|---|---|---|---|---|---|---|---|---|---|
| | | MSE | MAE | MSE | MAE | MSE | MAE | MSE | MAE | MSE | MAE | MSE | MAE | MSE | MAE |
| ETTh1 | 24 | **0.039** | **0.152** | 0.117 | 0.281 | 0.075 | 0.21 | 0.098 | 0.247 | 0.094 | 0.238 | 0.103 | 0.259 | **0.039** | 0.153 |
| | 48 | **0.062** | **0.191** | 0.192 | 0.369 | 0.227 | 0.402 | 0.158 | 0.319 | 0.21 | 0.367 | 0.167 | 0.328 | 0.064 | 0.196 |
| | 168 | 0.134 | 0.282 | 0.331 | 0.505 | 0.316 | 0.493 | 0.183 | 0.346 | 0.232 | 0.391 | 0.207 | 0.375 | **0.128** | **0.275** |
| | 336 | 0.154 | 0.31 | 0.353 | 0.525 | 0.306 | 0.495 | 0.222 | 0.387 | 0.232 | 0.388 | 0.23 | 0.398 | **0.15** | **0.303** |
| | 720 | 0.163 | 0.327 | 0.387 | 0.560 | 0.39 | 0.557 | 0.269 | 0.435 | 0.322 | 0.49 | 0.273 | 0.463 | **0.161** | **0.317** |
| ETTh2 | 24 | 0.090 | 0.229 | 0.106 | 0.255 | 0.103 | 0.249 | 0.093 | 0.240 | 0.198 | 0.345 | 0.102 | 0.255 | **0.086** | **0.217** |
| | 48 | 0.124 | 0.273 | 0.138 | 0.293 | 0.142 | 0.290 | 0.155 | 0.314 | 0.234 | 0.386 | 0.169 | 0.348 | **0.119** | **0.264** |
| | 168 | 0.208 | 0.360 | 0.211 | 0.368 | 0.227 | 0.376 | 0.232 | 0.389 | 0.331 | 0.453 | 0.246 | 0.422 | **0.185** | **0.339** |
| | 336 | 0.213 | 0.369 | 0.222 | 0.379 | 0.296 | 0.430 | 0.263 | 0.417 | 0.431 | 0.508 | 0.267 | 0.437 | **0.196** | **0.353** |
| | 720 | 0.214 | 0.374 | 0.238 | 0.394 | 0.325 | 0.463 | 0.277 | 0.431 | 0.437 | 0.517 | 0.303 | 0.493 | **0.209** | **0.370** |
| ETTm1 | 24 | 0.015 | 0.092 | 0.048 | 0.172 | 0.041 | 0.157 | 0.03 | 0.137 | 0.054 | 0.184 | 0.065 | 0.202 | **0.013** | **0.083** |
| | 48 | 0.027 | 0.126 | 0.076 | 0.219 | 0.101 | 0.257 | 0.069 | 0.203 | 0.190 | 0.361 | 0.078 | 0.220 | **0.025** | **0.116** |
| | 96 | 0.044 | 0.161 | 0.116 | 0.277 | 0.142 | 0.311 | 0.194 | 0.372 | 0.183 | 0.353 | 0.199 | 0.386 | **0.041** | **0.152** |
| | 288 | 0.103 | 0.246 | 0.233 | 0.413 | 0.318 | 0.472 | 0.401 | 0.554 | 0.186 | 0.362 | 0.411 | 0.572 | **0.092** | **0.231** |
| | 672 | 0.156 | 0.307 | 0.344 | 0.517 | 0.397 | 0.547 | 0.512 | 0.644 | 0.197 | 0.368 | 0.598 | 0.702 | **0.138** | **0.287** |
| Elec. | 24 | 0.260 | 0.288 | 0.261 | 0.297 | 0.263 | 0.279 | **0.251** | **0.275** | 0.427 | 0.330 | 0.528 | 0.447 | 0.256 | 0.288 |
| | 48 | 0.319 | 0.324 | **0.307** | 0.319 | 0.373 | 0.344 | 0.346 | 0.339 | 0.551 | 0.392 | 0.409 | 0.414 | **0.307** | **0.317** |
| | 168 | 0.427 | 0.394 | 0.438 | 0.403 | 0.609 | 0.462 | 0.544 | 0.424 | 0.893 | 0.538 | 0.959 | 0.612 | **0.426** | **0.391** |
| | 336 | 0.565 | 0.474 | 0.592 | 0.478 | 0.855 | 0.606 | 0.713 | 0.512 | 1.035 | 0.669 | 1.079 | 0.639 | **0.56** | **0.472** |
| | 720 | 0.861 | 0.643 | 0.885 | 0.663 | 1.263 | 0.858 | 1.182 | 0.806 | 1.548 | 0.881 | 1.001 | 0.714 | **0.859** | **0.638** |

Table 7: Multivariate time series forecasting results.

| Dataset | T | TS2Vec | | TS-TCC | | TNC | | Informer | | StemGNN | | LogTrans | | CoInception | |
|---|---|---|---|---|---|---|---|---|---|---|---|---|---|---|---|
| | | MSE | MAE | MSE | MAE | MSE | MAE | MSE | MAE | MSE | MAE | MSE | MAE | MSE | MAE |
| ETTh1 | 24 | 0.599 | 0.534 | 0.653 | 0.610 | 0.632 | 0.596 | 0.577 | 0.549 | 0.614 | 0.571 | 0.686 | 0.604 | **0.461** | **0.479** |
| | 48 | 0.629 | 0.555 | 0.720 | 0.693 | 0.705 | 0.688 | 0.685 | 0.625 | 0.748 | 0.618 | 0.766 | 0.757 | **0.512** | **0.503** |
| | 168 | 0.755 | 0.636 | 1.129 | 1.044 | 1.097 | 0.993 | 0.931 | 0.752 | **0.663** | **0.608** | 1.002 | 0.846 | 0.683 | **0.601** |
| | 336 | 0.907 | 0.717 | 1.492 | 1.076 | 1.454 | 0.919 | 1.128 | 0.873 | 0.927 | 0.730 | 1.362 | 0.952 | **0.829** | **0.678** |
| | 720 | 1.048 | 0.790 | 1.603 | 1.206 | 1.604 | 1.118 | 1.215 | 0.896 | - | - | 1.397 | 1.291 | **1.018** | **0.770** |
| ETTh2 | 24 | 0.398 | 0.461 | 0.883 | 0.747 | 0.830 | 0.756 | 0.720 | 0.665 | 1.292 | 0.883 | 0.828 | 0.750 | **0.335** | **0.432** |
| | 48 | 0.580 | 0.573 | 1.701 | 1.378 | 1.689 | 1.311 | 1.457 | 1.001 | 1.099 | 0.847 | 1.806 | 1.034 | **0.550** | **0.560** |
| | 168 | 1.901 | 1.065 | 3.956 | 2.301 | 3.792 | 2.029 | 3.489 | 1.515 | 2.282 | 1.228 | 4.070 | 1.681 | **1.812** | **1.055** |
| | 336 | 2.304 | 1.215 | 3.992 | 2.852 | 3.516 | 2.812 | 2.723 | 1.340 | 3.086 | 1.351 | 3.875 | 1.763 | **2.151** | **1.188** |
| | 720 | **2.650** | 1.373 | 4.732 | 2.345 | 4.501 | 2.410 | 3.467 | 1.473 | - | - | 3.913 | 1.552 | 2.962 | **1.338** |
| ETTm1 | 24 | 0.443 | 0.436 | 0.473 | 0.490 | 0.429 | 0.455 | **0.323** | **0.369** | 0.620 | 0.570 | 0.419 | 0.412 | 0.384 | 0.423 |
| | 48 | 0.582 | 0.515 | 0.671 | 0.665 | 0.623 | 0.602 | **0.494** | **0.503** | 0.744 | 0.628 | 0.507 | 0.583 | 0.552 | 0.521 |
| | 96 | 0.622 | 0.549 | 0.803 | 0.724 | 0.749 | 0.731 | 0.678 | 0.614 | 0.709 | 0.624 | 0.768 | 0.792 | **0.561** | **0.533** |
| | 288 | 0.709 | 0.609 | 1.958 | 1.429 | 1.791 | 1.356 | 1.056 | 0.786 | 0.843 | 0.683 | 1.462 | 1.320 | **0.623** | **0.578** |
| | 672 | 0.786 | 0.655 | 1.838 | 1.601 | 1.822 | 1.692 | 1.192 | 0.926 | - | - | 1.669 | 1.461 | **0.717** | **0.639** |
| Elec. | 24 | 0.287 | 0.374 | 0.278 | 0.370 | 0.305 | 0.384 | 0.312 | 0.387 | 0.439 | 0.388 | 0.297 | 0.374 | **0.234** | **0.335** |
| | 48 | 0.307 | 0.388 | 0.313 | 0.392 | 0.317 | 0.392 | 0.392 | 0.431 | 0.413 | 0.455 | 0.316 | 0.389 | **0.265** | **0.356** |
| | 168 | 0.332 | 0.407 | 0.338 | 0.411 | 0.358 | 0.423 | 0.515 | 0.509 | 0.506 | 0.518 | 0.426 | 0.466 | **0.282** | **0.372** |
| | 336 | 0.349 | 0.420 | 0.357 | 0.424 | 0.349 | 0.416 | 0.759 | 0.625 | 0.647 | 0.596 | 0.365 | 0.417 | **0.301** | **0.388** |
| | 720 | 0.375 | 0.438 | 0.382 | 0.442 | 0.447 | 0.486 | 0.969 | 0.788 | - | - | 0.344 | **0.403** | **0.331** | 0.409 |

## C.2 TIME SERIES CLASSIFICATION

**Additional details.** During the data processing stage, all datasets from the UCR Repository are normalized using z-score normalization, resulting in a mean of $0$ and a variance of $1$. Similarly, for datasets from the UEA Repository, each feature is independently normalized using z-score normalization. It is important to note that within the UCR Repository, there are three datasets that contain missing data points: *DodgerLoopDay*, *DodgerLoopGame*, and *DodgerLoopWeekend*. These datasets cannot be handled with T-Loss, TS-TCC, or TNC methods. However, with the employment of CoInception, we address this issue by directly replacing the missing values with $0$ and proceed with the training process as usual.

As stated in the main manuscript, the representations generated by CoInception are passed through a `MaxPooling` layer to extract the representative timestamp, which serves as the instance-level

representation of the input. This instance-level representation is subsequently utilized as the input for training the classifier. Consistent with Franceschi et al. (2019); Yue et al. (2022), we employ a Radial Basis Function (RBF) Support Vector Machine (SVM) classifier. The penalty parameter $C$ for the SVM is selected through a grid search conducted over the range $\{10^i | i \in [-4, 4]\}$.

**Additional results.** The comprehensive results of our CoInception framework on the 128 UCR Datasets, along with other baselines (TS2Vec Yue et al. (2022), T-Loss Franceschi et al. (2019), TS-TCC Eldele et al. (2021), TST Zerveas et al. (2021), TNC Tonekaboni et al. (2021), and DWT Chen et al. (2013)), are presented in Table 8. In general, CoInception outperforms other state-of-the-art methods in $67\%$ of the 128 datasets from the UCR Repository.

Similarly, detailed results for the 30 UEA Repository datasets are summarized in Table 9, accompanied by the corresponding Critical Difference diagram for the first 29 datasets depicted in Figure 8. In line with the findings in the univariate setting, CoInception also achieves better performance than more than $55\%$ of the datasets in the UEA Repository's multivariate scenario.

From both tables, it is evident that CoInception exhibits superior performance for the majority of datasets, resulting in a significant performance gap in terms of average accuracy.

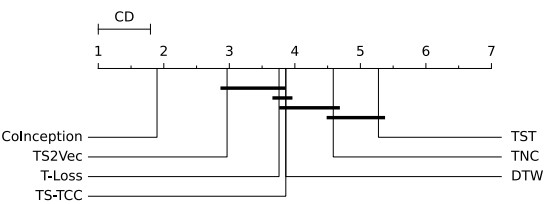

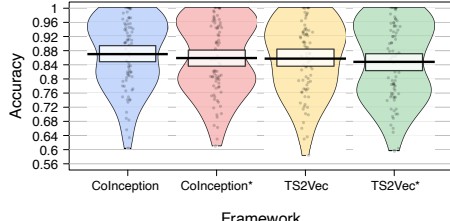

Figure 8: **Critical Difference Diagram**. Different classifiers' ranks on 29 Datasets from UEA Repository with the confidence level of $95\%$.

Figure 9: **Transferability Analysis**. Accuracy distribution for the first 85 UCR datasets.

### C.3    Time Series Abnormally Detection

**Additional details.** In the preprocessing stage, we utilize the Augmented Dickey-Fuller (ADF) test, as done in Tonekaboni et al. (2021); Yue et al. (2022), to determine the number of unit roots, denoted as $d$. Subsequently, the data is differenced $d$ times to mitigate any drifting effect, following the approach described in Yue et al. (2022).

For the evaluation process, we adopt a similar protocol as presented in Yue et al. (2022); Ren et al. (2019); Xu et al. (2018), aimed at relaxing the point-wise detection constraint. Within this protocol, a small delay is allowed after the appearance of each anomaly point. Specifically, for minutely data, a maximum delay of 7 steps is accepted, while for hourly data, a delay of 3 steps is employed. If the detector correctly identifies the point within this delay, all points within the corresponding segment are considered correct; otherwise, they are deemed incorrect.

### C.4    Noise Resiliency Techniques Analysis

In our current sampling strategy, DWT low-pass filter acts as a denoising method treating input series **x** as a signal. While an alternative of introducing noise (like jittering) to ensure *noise resiliency* is feasible, our preference for the DWT denoising technique stem from a realization. Jittering makes certain assumptions about the characteristics of the introduced noise, which may not be universally applicable to all time series or signals. In contrast, DWT denoising does not rely on such assumptions. The multiresolution breakdown in frequency achieved by DWT filters allows us to target specific high-frequency components prone to noise within the original signal. Through empirical analysis, we demonstrate the robustness of the DWT denoising technique compared to jittering.

We adhere to the commonly used parameters for the jittering augmentation technique described in Um et al. (2017); Iwana & Uchida (2021); Rashid & Louis (2019), where random noise is added from a Gaussian distribution with a mean ($\mu$) of 0 and a standard deviation ($\sigma$) of 0.03. To ensure a

Table 8: UCR 128 Datasets classification results. Best results are bold and highlighted in red.

| Dataset | TS2Vec | T-Loss | TNC | TS-TCC | TST | DTW | CoInception | CoInception's Rank |
|---|---|---|---|---|---|---|---|---|
| Adiac | 0.762 | 0.675 | 0.726 | **0.767** | 0.550 | 0.604 | **0.767** | 1 |
| ArrowHead | 0.857 | 0.766 | 0.703 | 0.737 | 0.771 | 0.703 | **0.863** | 1 |
| Beef | **0.767** | 0.667 | 0.733 | 0.600 | 0.500 | 0.633 | 0.733 | 2 |
| BeetleFly | 0.900 | 0.800 | 0.850 | 0.800 | **1.000** | 0.700 | 0.850 | 3 |
| BirdChicken | 0.800 | 0.850 | 0.750 | 0.650 | 0.650 | 0.750 | **0.900** | 1 |
| Car | 0.833 | 0.833 | 0.683 | 0.583 | 0.550 | 0.733 | **0.867** | 1 |
| CBF | **1.000** | 0.983 | 0.983 | 0.998 | 0.898 | 0.997 | **1.000** | 1 |
| ChlorineConcentration | **0.832** | 0.749 | 0.760 | 0.753 | 0.562 | 0.648 | 0.813 | 2 |
| CinCECGTorso | **0.827** | 0.713 | 0.669 | 0.671 | 0.508 | 0.651 | 0.765 | 2 |
| Coffee | **1.000** | **1.000** | **1.000** | **1.000** | 0.821 | **1.000** | **1.000** | 1 |
| Computers | 0.660 | 0.664 | 0.684 | **0.704** | 0.696 | 0.700 | 0.688 | 4 |
| CricketX | 0.782 | 0.713 | 0.623 | 0.731 | 0.385 | 0.754 | **0.805** | 1 |
| CricketY | 0.749 | 0.728 | 0.597 | 0.718 | 0.467 | 0.744 | **0.818** | 1 |
| CricketZ | 0.792 | 0.708 | 0.682 | 0.713 | 0.403 | 0.754 | **0.808** | 1 |
| DiatomSizeReduction | 0.984 | 0.984 | **0.993** | 0.977 | 0.961 | 0.967 | 0.984 | 2 |
| DistalPhalanxOutlineCorrect | 0.761 | 0.775 | 0.754 | 0.754 | 0.728 | 0.717 | **0.779** | 1 |
| DistalPhalanxOutlineAgeGroup | 0.727 | 0.727 | 0.741 | 0.755 | 0.741 | **0.770** | 0.748 | 3 |
| DistalPhalanxTW | 0.698 | 0.676 | 0.669 | 0.676 | 0.568 | 0.590 | **0.705** | 1 |
| Earthquakes | **0.748** | **0.748** | **0.748** | **0.748** | **0.748** | 0.719 | **0.748** | 1 |
| ECG200 | 0.920 | **0.940** | 0.830 | 0.880 | 0.830 | 0.770 | 0.920 | 2 |
| ECG5000 | 0.935 | 0.933 | 0.937 | 0.941 | 0.928 | 0.924 | **0.944** | 1 |
| ECGFiveDays | **1.000** | **1.000** | 0.999 | 0.878 | 0.763 | 0.768 | **1.000** | 1 |
| ElectricDevices | 0.721 | 0.707 | 0.700 | 0.686 | 0.676 | 0.602 | **0.741** | 1 |
| FaceAll | 0.771 | 0.786 | 0.766 | 0.813 | 0.504 | 0.808 | **0.842** | 1 |
| FaceFour | 0.932 | 0.920 | 0.659 | 0.773 | 0.511 | 0.830 | **0.955** | 1 |
| FacesUCR | 0.924 | 0.884 | 0.789 | 0.863 | 0.543 | 0.905 | **0.928** | 1 |
| FiftyWords | 0.771 | 0.732 | 0.653 | 0.653 | 0.525 | 0.690 | **0.778** | 1 |
| Fish | 0.926 | 0.891 | 0.817 | 0.817 | 0.720 | 0.823 | **0.954** | 1 |
| FordA | **0.936** | 0.928 | 0.902 | 0.930 | 0.568 | 0.555 | 0.930 | 2 |
| FordB | 0.794 | 0.793 | 0.733 | 0.815 | 0.507 | 0.620 | **0.832** | 1 |
| GunPoint | 0.980 | 0.980 | 0.967 | **0.993** | 0.827 | 0.907 | 0.987 | 2 |
| Ham | 0.714 | 0.724 | 0.752 | 0.743 | 0.524 | 0.467 | **0.810** | 1 |
| HandOutlines | 0.922 | 0.922 | 0.930 | 0.724 | 0.735 | 0.881 | **0.935** | 1 |
| Haptics | **0.526** | 0.490 | 0.474 | 0.396 | 0.357 | 0.377 | 0.510 | 2 |
| Herring | **0.641** | 0.594 | 0.594 | 0.594 | 0.594 | 0.531 | 0.594 | 2 |
| InlineSkate | 0.415 | 0.371 | 0.378 | 0.347 | 0.287 | 0.384 | **0.424** | 1 |
| InsectWingbeatSound | 0.630 | 0.597 | 0.549 | 0.415 | 0.266 | 0.355 | **0.634** | 1 |
| ItalyPowerDemand | 0.925 | 0.954 | 0.928 | 0.955 | 0.845 | 0.950 | **0.962** | 1 |
| LargeKitchenAppliances | 0.845 | 0.789 | 0.776 | 0.848 | 0.595 | 0.795 | **0.893** | 1 |
| Lightning2 | 0.869 | 0.869 | 0.869 | 0.836 | 0.705 | 0.869 | **0.902** | 1 |
| Lightning7 | **0.863** | 0.795 | 0.767 | 0.685 | 0.411 | 0.726 | 0.836 | 2 |
| Mallat | 0.914 | 0.951 | 0.871 | 0.922 | 0.713 | 0.934 | **0.953** | 1 |
| Meat | 0.950 | 0.950 | 0.917 | 0.883 | 0.900 | 0.933 | **0.967** | 1 |
| MedicalImages | 0.789 | 0.750 | 0.754 | 0.747 | 0.632 | 0.737 | **0.795** | 1 |
| MiddlePhalanxOutlineCorrect | **0.838** | 0.825 | 0.818 | 0.818 | 0.753 | 0.698 | 0.832 | 2 |
| MiddlePhalanxOutlineAgeGroup | 0.636 | **0.656** | 0.643 | 0.630 | 0.617 | 0.500 | **0.656** | 1 |
| MiddlePhalanxTW | 0.584 | 0.591 | 0.571 | **0.610** | 0.506 | 0.506 | 0.604 | 2 |
| MoteStrain | 0.861 | 0.851 | 0.825 | 0.843 | 0.768 | 0.835 | **0.873** | 1 |
| NonInvasiveFetalECGThorax1 | **0.930** | 0.878 | 0.898 | 0.898 | 0.471 | 0.790 | 0.919 | 2 |
| NonInvasiveFetalECGThorax2 | 0.938 | 0.919 | 0.912 | 0.913 | 0.832 | 0.865 | **0.942** | 1 |
| OliveOil | **0.900** | 0.867 | 0.833 | 0.800 | 0.800 | 0.833 | **0.900** | 1 |
| OSULeaf | **0.851** | 0.760 | 0.723 | 0.723 | 0.545 | 0.591 | 0.835 | 2 |
| PhalangesOutlinesCorrect | 0.809 | 0.784 | 0.787 | 0.804 | 0.773 | 0.728 | **0.818** | 1 |
| Phoneme | **0.312** | 0.276 | 0.180 | 0.242 | 0.139 | 0.228 | 0.310 | 2 |
| Plane | **1.000** | 0.990 | **1.000** | **1.000** | 0.933 | **1.000** | **1.000** | 1 |
| ProximalPhalanxOutlineCorrect | 0.887 | 0.859 | 0.866 | 0.873 | 0.770 | 0.784 | **0.911** | 1 |
| ProximalPhalanxOutlineAgeGroup | 0.834 | 0.844 | **0.854** | 0.839 | **0.854** | 0.805 | 0.849 | 3 |
| ProximalPhalanxTW | **0.824** | 0.771 | 0.810 | 0.800 | 0.780 | 0.761 | **0.824** | 1 |

fair evaluation, we adopt all the settings as CoInception framework, making alterations only to the strategy employed in generating the perturbed series $\mathbf{x}$ during the training process.

Our experiments encompass all three main tasks, and the full results are reported in Table 10. Regarding the classification task, we present average performance metrics across a set of 5 UCR datasets and 5 UEA datasets. 5 datasets in UCR repository include *Rock, PigCVP, CinCECGTorso, SemgHand-MovementCh2, HouseTwenty*; while the chosen datasets from UEA repository are *DuckDuckGeese, AtrialFibrillation, Handwriting, RacketSports, SelfRegulationSCP1*. In the context of the forecasting task, we execute univariate experiments utilizing the ETTm1 dataset, with the results averaged across various prediction horizons, encompassing both short-term and long-term forecasts. As for the anomaly detection task, we offer scores for the Yahoo dataset under normal circumstances. Across these three tasks, the DWT-based denoising technique consistently demonstrates its notable superior-

| Dataset | TS2Vec | T-Loss | TNC | TS-TCC | TST | DTW | CoInception | CoInception's Rank |
|---|---|---|---|---|---|---|---|---|
| RefrigerationDevices | 0.589 | 0.515 | 0.565 | 0.563 | 0.483 | 0.464 | **0.597** | 1 |
| ScreenType | 0.411 | 0.416 | **0.509** | 0.419 | 0.419 | 0.397 | 0.413 | 5 |
| ShapeletSim | **1.000** | 0.672 | 0.589 | 0.683 | 0.489 | 0.650 | 0.994 | 2 |
| ShapesAll | **0.902** | 0.848 | 0.788 | 0.773 | 0.733 | 0.768 | 0.898 | 2 |
| SmallKitchenAppliances | 0.731 | 0.677 | 0.725 | 0.691 | 0.592 | 0.643 | **0.792** | 1 |
| SonyAIBORobotSurface1 | 0.903 | 0.902 | 0.804 | 0.899 | 0.724 | 0.725 | **0.908** | 1 |
| SonyAIBORobotSurface2 | 0.871 | 0.889 | 0.834 | 0.907 | 0.745 | 0.831 | **0.939** | 1 |
| StarLightCurves | 0.969 | 0.964 | 0.968 | 0.967 | 0.949 | 0.907 | **0.971** | 1 |
| Strawberry | 0.962 | 0.954 | 0.951 | 0.965 | 0.916 | 0.941 | **0.970** | 1 |
| SwedishLeaf | 0.941 | 0.914 | 0.880 | 0.923 | 0.738 | 0.792 | **0.950** | 1 |
| Symbols | **0.976** | 0.963 | 0.885 | 0.916 | 0.786 | 0.950 | 0.970 | 2 |
| SyntheticControl | 0.997 | 0.987 | **1.000** | 0.990 | 0.490 | 0.993 | 0.997 | 2 |
| ToeSegmentation1 | 0.917 | 0.939 | 0.864 | 0.930 | 0.807 | 0.772 | **0.943** | 1 |
| ToeSegmentation2 | 0.892 | 0.900 | 0.831 | 0.877 | 0.615 | 0.838 | **0.908** | 1 |
| Trace | **1.000** | 0.990 | **1.000** | **1.000** | **1.000** | **1.000** | **1.000** | 1 |
| TwoLeadECG | 0.986 | **0.999** | 0.993 | 0.976 | 0.871 | 0.905 | 0.998 | 2 |
| TwoPatterns | **1.000** | 0.999 | **1.000** | 0.999 | 0.466 | **1.000** | **1.000** | 1 |
| UWaveGestureLibraryX | 0.795 | 0.785 | 0.781 | 0.733 | 0.569 | 0.728 | **0.817** | 1 |
| UWaveGestureLibraryY | 0.719 | 0.710 | 0.697 | 0.641 | 0.348 | 0.634 | **0.739** | 1 |
| UWaveGestureLibraryZ | 0.770 | 0.757 | 0.721 | 0.690 | 0.655 | 0.658 | **0.771** | 1 |
| UWaveGestureLibraryAll | 0.930 | 0.896 | 0.903 | 0.692 | 0.475 | 0.892 | **0.937** | 1 |
| Wafer | 0.998 | 0.992 | 0.994 | 0.994 | 0.991 | 0.980 | **0.999** | 1 |
| Wine | 0.870 | 0.815 | 0.759 | 0.778 | 0.500 | 0.574 | **0.907** | 1 |
| WordSynonyms | 0.676 | **0.691** | 0.630 | 0.531 | 0.422 | 0.649 | 0.683 | 2 |
| Worms | 0.701 | 0.727 | 0.623 | **0.753** | 0.455 | 0.584 | 0.740 | 2 |
| WormsTwoClass | 0.805 | 0.792 | 0.727 | 0.753 | 0.584 | 0.623 | **0.818** | 1 |
| Yoga | **0.887** | 0.837 | 0.812 | 0.791 | 0.830 | 0.837 | 0.882 | 2 |
| ACSF1 | 0.900 | 0.900 | 0.730 | 0.730 | 0.760 | 0.640 | **0.910** | 1 |
| AllGestureWiimoteX | 0.777 | 0.763 | 0.703 | 0.697 | 0.259 | 0.716 | **0.799** | 1 |
| AllGestureWiimoteY | **0.793** | 0.726 | 0.699 | 0.741 | 0.423 | 0.729 | 0.776 | 2 |
| AllGestureWiimoteZ | 0.746 | 0.723 | 0.646 | 0.689 | 0.447 | 0.643 | **0.747** | 1 |
| BME | **0.993** | **0.993** | 0.973 | 0.933 | 0.760 | 0.900 | 0.980 | 3 |
| Chinatown | 0.965 | 0.951 | 0.977 | 0.983 | 0.936 | 0.957 | **0.985** | 1 |
| Crop | 0.756 | 0.722 | 0.738 | 0.742 | 0.710 | 0.665 | **0.757** | 1 |
| EOGHorizontalSignal | 0.539 | **0.605** | 0.442 | 0.401 | 0.373 | 0.503 | 0.577 | 2 |
| EOGVerticalSignal | 0.503 | 0.434 | 0.392 | 0.376 | 0.298 | 0.448 | **0.564** | 1 |
| EthanolLevel | 0.468 | 0.382 | 0.424 | 0.486 | 0.260 | 0.276 | **0.496** | 1 |
| FreezerRegularTrain | 0.986 | 0.956 | 0.991 | 0.989 | 0.922 | 0.899 | **0.994** | 1 |
| FreezerSmallTrain | 0.870 | 0.933 | **0.982** | 0.979 | 0.920 | 0.753 | 0.919 | 5 |
| Fungi | 0.957 | **1.000** | 0.527 | 0.753 | 0.366 | 0.839 | 0.962 | 2 |
| GestureMidAirD1 | 0.608 | 0.608 | 0.431 | 0.369 | 0.208 | 0.569 | **0.662** | 1 |
| GestureMidAirD2 | 0.469 | 0.546 | 0.362 | 0.254 | 0.138 | **0.608** | 0.592 | 2 |
| GestureMidAirD3 | 0.292 | 0.285 | 0.292 | 0.177 | 0.154 | 0.323 | **0.392** | 1 |
| GesturePebbleZ1 | **0.930** | 0.919 | 0.378 | 0.395 | 0.500 | 0.791 | 0.872 | 3 |
| GesturePebbleZ2 | 0.873 | 0.899 | 0.316 | 0.430 | 0.380 | 0.671 | **0.911** | 1 |
| GunPointAgeSpan | 0.987 | 0.994 | 0.984 | 0.994 | 0.991 | 0.918 | **1.000** | 1 |
| GunPointMaleVersusFemale | **1.000** | 0.997 | 0.994 | 0.997 | **1.000** | 0.997 | **1.000** | 1 |
| GunPointOldVersusYoung | **1.000** | **1.000** | **1.000** | **1.000** | **1.000** | 0.838 | **1.000** | 1 |
| HouseTwenty | 0.916 | **0.933** | 0.782 | 0.790 | 0.815 | 0.924 | 0.899 | 4 |
| InsectEPGRegularTrain | **1.000** | **1.000** | **1.000** | **1.000** | **1.000** | 0.872 | **1.000** | 1 |
| InsectEPGSmallTrain | **1.000** | **1.000** | **1.000** | **1.000** | **1.000** | 0.735 | **1.000** | 1 |
| MelbournePedestrian | 0.959 | 0.944 | 0.942 | 0.949 | 0.741 | 0.791 | **0.961** | 1 |
| MixedShapesRegularTrain | 0.917 | 0.905 | 0.911 | 0.855 | 0.879 | 0.842 | **0.933** | 1 |
| MixedShapesSmallTrain | 0.861 | 0.860 | 0.813 | 0.735 | 0.828 | 0.780 | **0.876** | 1 |
| PickupGestureWiimoteZ | 0.820 | 0.740 | 0.620 | 0.600 | 0.240 | 0.660 | **0.880** | 1 |
| PigAirwayPressure | 0.630 | 0.510 | 0.413 | 0.380 | 0.120 | 0.106 | **0.827** | 1 |
| PigArtPressure | **0.966** | 0.928 | 0.808 | 0.524 | 0.774 | 0.245 | **0.966** | 1 |
| PigCVP | 0.812 | 0.788 | 0.649 | 0.615 | 0.596 | 0.154 | **0.899** | 1 |
| PLAID | 0.561 | 0.555 | 0.495 | 0.445 | 0.419 | **0.840** | 0.533 | 4 |
| PowerCons | 0.961 | 0.900 | 0.933 | 0.961 | 0.911 | 0.878 | **0.983** | 1 |

ity over the jittering technique. It's worth reiterating that jittering assumes specific characteristics of the introduced noise, which may not universally apply to all time series or signals. In contrast, DWT denoising relies on an assumption generally applicable to natural signals: noisy elements typically manifest as high-frequency components within the original signal. We leave theoretical analysis and further exploration as open questions for our future research.

## C.5 RECEPTIVE FIELD ANALYSIS

This experiment aims to investigate the scalability of the CoInception framework in comparison to the stacked Dilated Convolution network proposed in Yue et al. (2022). We present a visualization of the relationship between the network depth, the number of parameters, and the maximum receptive fields of output timestamps in Figure 10.

| Dataset | TS2Vec | T-Loss | TNC | TS-TCC | TST | DTW | CoInception | CoInception's Rank |
|---|---|---|---|---|---|---|---|---|
| Rock | **0.700** | 0.580 | 0.580 | 0.600 | 0.680 | 0.600 | 0.660 | 3 |
| SemgHandGenderCh2 | **0.963** | 0.890 | 0.882 | 0.837 | 0.725 | 0.802 | 0.962 | 2 |
| SemgHandMovementCh2 | **0.860** | 0.789 | 0.593 | 0.613 | 0.420 | 0.584 | 0.811 | 2 |
| SemgHandSubjectCh2 | **0.951** | 0.853 | 0.771 | 0.753 | 0.484 | 0.727 | 0.918 | 2 |
| ShakeGestureWiimoteZ | **0.940** | 0.920 | 0.820 | 0.860 | 0.760 | 0.860 | 0.920 | 2 |
| SmoothSubspace | 0.980 | 0.960 | 0.913 | 0.953 | 0.827 | 0.827 | **0.993** | 1 |
| UMD | **1.000** | 0.993 | 0.993 | 0.986 | 0.910 | 0.993 | **1.000** | 1 |
| DodgerLoopDay | 0.562 | – | – | – | 0.200 | 0.500 | **0.588** | 1 |
| DodgerLoopGame | 0.841 | – | – | – | 0.696 | 0.877 | **0.884** | 1 |
| DodgerLoopWeekend | 0.964 | – | – | – | 0.732 | 0.949 | **0.986** | 1 |
| Avg. (first 125 datasets) | 0.829 | 0.806 | 0.761 | 0.757 | 0.641 | 0.726 | **0.843** | **1** |

Table 9: UEA 30 Datasets classification results. Best results are bold and highlighted in **red**.

| Dataset | TS2Vec | T-Loss | TNC | TS-TCC | TST | DTW | CoInception | Rank |
|---|---|---|---|---|---|---|---|---|
| ArticularyWordRecognition | **0.987** | 0.943 | 0.973 | 0.953 | 0.977 | **0.987** | **0.987** | 1 |
| AtrialFibrillation | 0.200 | 0.133 | 0.133 | 0.267 | 0.067 | 0.200 | **0.333** | 1 |
| BasicMotions | 0.975 | **1.000** | 0.975 | **1.000** | 0.975 | 0.975 | **1.000** | 1 |
| CharacterTrajectories | **0.995** | 0.993 | 0.967 | 0.985 | 0.975 | 0.989 | 0.992 | 3 |
| Cricket | 0.972 | 0.972 | 0.958 | 0.917 | **1.000** | **1.000** | 0.986 | 3 |
| DuckDuckGeese | **0.680** | 0.650 | 0.460 | 0.380 | 0.620 | 0.600 | 0.500 | 5 |
| EigenWorms | **0.847** | 0.840 | 0.840 | 0.779 | 0.748 | 0.618 | **0.847** | 1 |
| Epilepsy | 0.964 | 0.971 | 0.957 | 0.957 | 0.949 | 0.964 | **0.978** | 1 |
| ERing | 0.874 | 0.133 | 0.852 | **0.904** | 0.874 | 0.133 | 0.900 | 2 |
| EthanolConcentration | 0.308 | 0.205 | 0.297 | 0.285 | 0.262 | **0.323** | 0.319 | 2 |
| FaceDetection | 0.501 | 0.513 | 0.536 | 0.544 | 0.534 | 0.529 | **0.550** | 1 |
| FingerMovements | 0.480 | **0.580** | 0.470 | 0.460 | 0.560 | 0.530 | 0.550 | 3 |
| HandMovementDirection | 0.338 | **0.351** | 0.324 | 0.243 | 0.243 | 0.231 | **0.351** | 1 |
| Handwriting | 0.515 | 0.451 | 0.249 | 0.498 | 0.225 | 0.286 | **0.549** | 1 |
| Heartbeat | 0.683 | 0.741 | 0.746 | 0.751 | 0.746 | 0.717 | **0.790** | 1 |
| JapaneseVowels | 0.984 | 0.989 | 0.978 | 0.930 | 0.978 | 0.949 | **0.992** | 1 |
| Libras | 0.867 | **0.883** | 0.817 | 0.822 | 0.656 | 0.870 | 0.867 | 3 |
| LSST | 0.537 | 0.509 | **0.595** | 0.474 | 0.408 | 0.551 | 0.537 | 3 |
| MotorImagery | 0.510 | 0.580 | 0.500 | **0.610** | 0.500 | 0.500 | 0.560 | 3 |
| NATOPS | 0.928 | 0.917 | 0.911 | 0.822 | 0.850 | 0.883 | **0.972** | 1 |
| PEMS-SF | 0.682 | 0.676 | 0.699 | 0.734 | 0.740 | 0.711 | **0.786** | 1 |
| PenDigits | 0.989 | 0.981 | 0.979 | 0.974 | 0.560 | 0.977 | **0.991** | 1 |
| PhonemeSpectra | 0.233 | 0.222 | 0.207 | 0.252 | 0.085 | 0.151 | **0.260** | 1 |
| RacketSports | 0.855 | 0.855 | 0.776 | 0.816 | 0.809 | 0.803 | **0.868** | 1 |
| SelfRegulationSCP1 | 0.812 | **0.843** | 0.799 | 0.823 | 0.754 | 0.775 | 0.765 | 6 |
| SelfRegulationSCP2 | **0.578** | 0.539 | 0.550 | 0.533 | 0.550 | 0.539 | 0.556 | 2 |
| SpokenArabicDigits | **0.988** | 0.905 | 0.934 | 0.970 | 0.923 | 0.963 | 0.979 | 2 |
| StandWalkJump | 0.467 | 0.333 | 0.400 | 0.333 | 0.267 | 0.200 | **0.533** | 1 |
| UWaveGestureLibrary | **0.906** | 0.875 | 0.759 | 0.753 | 0.575 | 0.903 | 0.894 | 3 |
| InsectWingbeat | 0.466 | 0.156 | **0.469** | 0.264 | 0.105 | – | 0.449 | 3 |
| Avg. (first 29 datasets) | 0.712 | 0.675 | 0.677 | 0.682 | 0.635 | 0.650 | 0.731 | **1** |

The receptive field represents the number of input timestamps involved in calculating an output timestamp. The reported statistics for both the number of parameters and the receptive field are presented in logarithmic scale to ensure smoothness and a smaller number range.

As depicted in the figure, CoInception consistently exhibits a lower number of parameters compared to TS2Vec, across a network depth ranging from 1 to 30 layers. It is worth noting that the inclusion of a 30-layer CoInception framework in the visualization is purely for illustrative purposes, as we believe a much smaller depth is sufficient for the majority of time series datasets. In fact, we only utilize 3 layers for all datasets in the remaining sections. Furthermore, CoInception, with its multiple Basic units of varying filter lengths, can easily achieve very large receptive fields even with just a few layers.

## C.6    CLUSTERABILITY ANALYSIS

Through this experiment, we test the clusterability of the learnt representations in the latent space. We visualize the feature representations with t-SNE proposed by Maaten and partners - van der Maaten & Hinton (2008) in two dimensional space. In the best scenario, the representations should be presented in latent space by groups of clusters, basing on their labels - their underlying states.

Table 10: Noise Resiliency Techniques Comparison

| Task | | CoInception w. jittering | CoInception w. DWT filtering |
|---|---|---|---|
| Classification | Acc. | 0.656 (- 6.95%) | **0.705** |
| | AUC. | 0.727 (- 6.07%) | **0.774** |
| Forecasting | MSE | 0.12 (- 49.12%) | **0.061** |
| | MAE | 0.262 (- 33.91%) | **0.173** |
| Anomaly Detection | F1 | 0.613 (- 20.28%) | **0.769** |
| | P. | 0.548 (- 30.63%) | **0.790** |
| | R. | 0.695 (- 7.08%) | **0.748** |

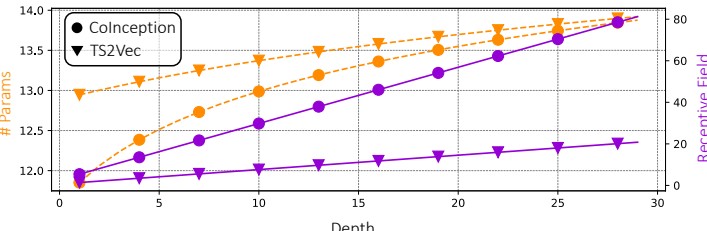

Figure 10: **Receptive Field Analysis**. The relation between models' depth with their number of parameters and their maximum receptive field.

Figure 11 compares the distribution of representations learned by CoInception and TS2Vec in three dataset with greatest test set in UCR 128 repository. It is evident that the proposed CoInception does outperform the second best TS2Vec in terms of representation learning from the same hidden state. The clusters learnt by CoInception are more compact than those produced by TS2Vec, especially when the number of classes increase for *ElectricDevices* or *Crop* datasets.

## C.7 TRANSFERABILITY ANALYSIS

We assess the transferability of CoInception framework under all three tasks: forecasting, classification and anomaly detection.

For the forecasting task, we evaluate the transferability of the CoInception framework using the following approach. The ETT datasets Zhou et al. (2021) consist of power transformer data collected from July 2016 to July 2018. We focus on the small datasets, which include data from 2 stations, specifically load and oil temperature. ETTh1 and ETTh2 are datasets with a temporal granularity of 1 hour, corresponding to the two stations. Since these two datasets exhibit high correlation, we leverage transfer learning between them. Initially, we perform the unsupervised learning step on the ETTh1 dataset, similar to the process used for forecasting assessment. Subsequently, the weights of the CoInception Encoder are frozen, and we utilize this pre-trained Encoder for training the forecasting framework, employing a Ridge Regression model, on the ETTh2 dataset. The detailed results are

Table 11: Transferability analysis with time series forecasting task.

| Model | Forecasting (ETTh1 ->ETTh2) | | | | |
|---|---|---|---|---|---|
| | 24 Step | 48 Step | 168 Step | 336 Step | 720 Step |
| TS2Vec | 0.090 | 0.124 | 0.208 | 0.213 | 0.214 |
| TS2Vec* | 0.100 | 0.143 | 0.236 | 0.223 | 0.217 |
| CoInception | 0.086 | 0.119 | **0.185** | **0.196** | **0.209** |
| CoInception* | **0.084** | **0.118** | 0.188 | 0.201 | 0.211 |

presented in Table 11. Overall, CoInception demonstrates its strong adaptability to the ETTh2 dataset, surpassing TS2Vec and even performing comparably to its own results in the regular forecasting setting.

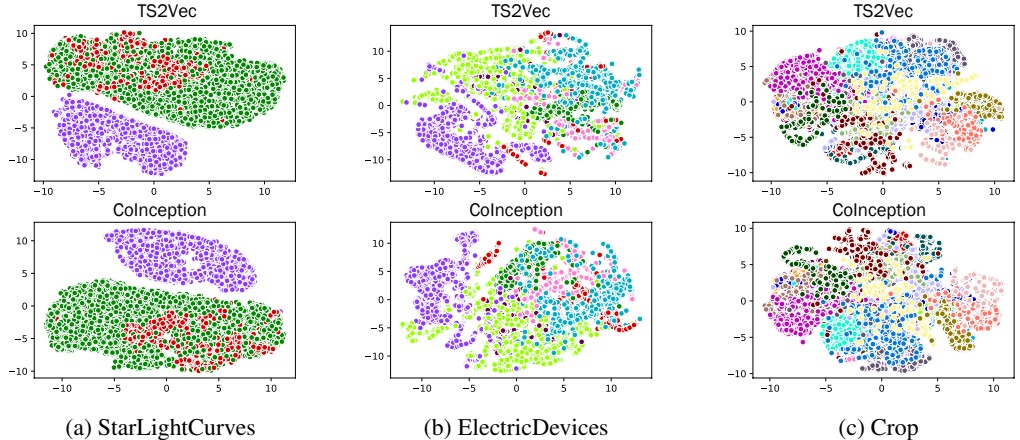

|  | TS2Vec | TS2Vec | TS2Vec |
|---|---|---|---|
|  | CoInception | CoInception | CoInception |
|  | (a) StarLightCurves | (b) ElectricDevices | (c) Crop |

Figure 11: Comparing clusterability of our CoInception with TS2Vec over three benchmark datasets in UCR 125 Repository.

Figure 12: Ablation analysis for Inception block of CoInception framework.

|  | CoInception (2a) | CoInception (2b) | CoInception (2c) | **CoInception** |
|---|---|---|---|---|
| *Classification:* | | | | |
| Acc. | 0.671 (- 4.82%) | 0.571 (- 19.00%) | 0.654 (- 7.23%) | 0.705 |
| AUC. | 0.734 (- 5.16%) | 0.635 (- 17.95%) | 0.719 (- 7.11%) | 0.774 |
| *Forecasting:* | | | | |
| MSE | 0.068 (- 10.29%) | 0.064 (- 4.68%) | 0.060 (+ 1.66%) | 0.061 |
| MAE | 0.186 (- 6.98%) | 0.178 (- 2.80%) | 0.172 (+ 0.58%) | 0.173 |
| *Anomaly Detection:* | | | | |
| F1 | 0.648 (- 15.73%) | 0.647 (- 15.86%) | 0.653 (-15.08%) | 0.769 |
| P. | 0.626 (- 20.75%) | 0.639 (- 19.11%) | 0.617 (-21.89%) | 0.790 |
| R. | 0.671 (- 10.29%) | 0.656 (- 12.29%) | 0.694 (- 7.21%) | 0.748 |

For classification task, we follow the settings in Franceschi et al. (2019). We first train our Encoder unsupervisedly with training data from FordA dataset. Following, for each dataset in UCR repository, the SVM classifier is trained on top of the representations produced by the frozen CoInception Encoder with this dataset. Table 12 provides a summary of the transferability results on the first 85 UCR datasets. Although CoInception exhibits lower performance compared to its own results in the regular classification setting in most datasets, its overall performance, as measured by the average accuracy, is still comparable to TS2Vec in its normal settings.

For anomaly detection, the settings are inherited from Ren et al. (2019); Yue et al. (2022), and we have already presented the results with cold-start settings in the main manuscript.

## C.8 ADDITIONAL ABLATION ANALYSIS

Through this experiment, we further analyze the effect of each individual contribution within Inception block. To be specific, three variances are adopts: **(2a)** We replace Aggregator with simple concatenation operation, and add Bottleneck layer followed the design in Wu et al. (2022); **(2b)** Dilated Convolution are turned into normal 1D Convolution layer; **(2c)** Skip connections between Basic Units of different layers are removed. The results are provided in Table 12. Overall, while different ablations show the greater detrimental levels in different tasks, we consistently notice a decline in the whole performance when any suggested alteration is excluded or substituted. This pattern indicates the positive impact of each change on the robustness of the CoInception framework.

Table 12: Transferability analysis for time series classification.

| Dataset | TS2Vec | TS2Vec* | T-Loss | T-Loss* | CoInception | CoInception* |
|---|---|---|---|---|---|---|
| Adiac | 0.762 | 0.783 | 0.760 | 0.716 | 0.767 | **0.803** |
| ArrowHead | 0.857 | 0.829 | 0.817 | 0.829 | **0.863** | 0.806 |
| Beef | **0.767** | 0.700 | 0.667 | 0.700 | 0.733 | 0.733 |
| BeetleFly | **0.900** | **0.900** | 0.800 | **0.900** | 0.850 | **0.900** |
| BirdChicken | 0.800 | 0.800 | **0.900** | 0.800 | **0.900** | 0.800 |
| Car | 0.833 | 0.817 | 0.850 | 0.817 | 0.867 | **0.883** |
| CBF | **1.000** | **1.000** | 0.988 | 0.994 | **1.000** | 0.997 |
| ChlorineConcentration | **0.832** | 0.802 | 0.688 | 0.782 | 0.813 | 0.814 |
| CinCECGTorso | **0.827** | 0.738 | 0.638 | 0.740 | 0.765 | 0.772 |
| Coffee | **1.000** | **1.000** | **1.000** | **1.000** | **1.000** | **1.000** |
| Computers | 0.660 | 0.660 | 0.648 | 0.628 | **0.688** | 0.668 |
| CricketX | 0.782 | 0.767 | 0.682 | 0.777 | **0.805** | 0.767 |
| CricketY | 0.749 | 0.746 | 0.667 | 0.767 | **0.818** | 0.751 |
| CricketZ | 0.792 | 0.772 | 0.656 | 0.764 | **0.808** | 0.762 |
| DiatomSizeReduction | 0.984 | 0.961 | 0.974 | **0.993** | 0.984 | 0.977 |
| DistalPhalanxOutlineCorrect | 0.761 | 0.757 | 0.764 | 0.768 | **0.779** | 0.775 |
| DistalPhalanxOutlineAgeGroup | 0.727 | **0.748** | 0.727 | 0.734 | **0.748** | 0.741 |
| DistalPhalanxTW | 0.698 | 0.669 | 0.669 | 0.676 | **0.705** | 0.698 |
| Earthquakes | **0.748** | **0.748** | **0.748** | **0.748** | **0.748** | **0.748** |
| ECG200 | **0.920** | 0.910 | 0.830 | 0.900 | **0.920** | **0.920** |
| ECG5000 | 0.935 | 0.935 | 0.940 | 0.936 | **0.944** | 0.942 |
| ECGFiveDays | **1.000** | **1.000** | **1.000** | **1.000** | **1.000** | **1.000** |
| ElectricDevices | 0.721 | 0.714 | 0.676 | 0.732 | **0.741** | 0.722 |
| FaceAll | 0.771 | 0.786 | 0.734 | 0.802 | **0.842** | 0.821 |
| FaceFour | 0.932 | 0.898 | 0.830 | 0.875 | **0.955** | 0.807 |
| FacesUCR | 0.924 | **0.928** | 0.835 | 0.918 | **0.928** | 0.923 |
| FiftyWords | 0.771 | 0.785 | 0.745 | 0.780 | 0.778 | **0.804** |
| Fish | 0.926 | 0.949 | **0.960** | 0.880 | 0.954 | 0.943 |
| FordA | 0.936 | 0.936 | 0.927 | 0.935 | 0.930 | **0.943** |
| FordB | 0.794 | 0.779 | 0.798 | **0.810** | 0.802 | 0.796 |
| GunPoint | 0.980 | **0.993** | 0.987 | **0.993** | 0.987 | 0.987 |
| Ham | 0.714 | 0.714 | 0.533 | 0.695 | **0.810** | 0.648 |
| HandOutlines | 0.922 | 0.919 | 0.919 | 0.922 | **0.935** | 0.930 |
| Haptics | **0.526** | **0.526** | 0.474 | 0.455 | 0.510 | 0.513 |
| Herring | **0.641** | 0.594 | 0.578 | 0.578 | 0.594 | 0.609 |
| InlineSkate | 0.415 | **0.465** | 0.444 | 0.447 | 0.424 | 0.453 |
| InsectWingbeatSound | 0.630 | 0.603 | 0.599 | 0.623 | **0.634** | 0.630 |
| ItalyPowerDemand | 0.925 | 0.957 | 0.929 | 0.925 | 0.962 | **0.963** |
| LargeKitchenAppliances | 0.845 | 0.861 | 0.765 | 0.848 | **0.893** | 0.787 |
| Lightning2 | 0.869 | **0.918** | 0.787 | **0.918** | 0.902 | 0.852 |
| Lightning7 | **0.863** | 0.781 | 0.740 | 0.795 | 0.836 | 0.808 |
| Mallat | 0.914 | 0.956 | 0.916 | 0.964 | 0.953 | **0.966** |
| Meat | 0.950 | **0.967** | 0.867 | 0.950 | **0.967** | **0.967** |
| MedicalImages | 0.789 | 0.784 | 0.725 | 0.784 | **0.795** | 0.792 |
| MiddlePhalanxOutlineCorrect | **0.838** | 0.794 | 0.787 | 0.814 | 0.832 | **0.838** |
| MiddlePhalanxOutlineAgeGroup | 0.636 | 0.649 | 0.623 | 0.656 | 0.656 | **0.662** |
| MiddlePhalanxTW | 0.584 | 0.597 | 0.584 | **0.610** | 0.604 | **0.610** |
| MoteStrain | 0.861 | 0.847 | 0.823 | 0.871 | **0.873** | 0.822 |
| NonInvasiveFetalECGThorax1 | 0.930 | 0.946 | 0.925 | 0.910 | 0.919 | **0.947** |
| NonInvasiveFetalECGThorax2 | 0.938 | **0.955** | 0.930 | 0.927 | 0.942 | 0.950 |
| OliveOil | **0.900** | **0.900** | **0.900** | **0.900** | **0.900** | **0.900** |
| OSULeaf | 0.851 | **0.868** | 0.736 | 0.831 | 0.835 | 0.777 |
| PhalangesOutlinesCorrect | 0.809 | 0.794 | 0.784 | 0.801 | **0.818** | 0.800 |
| Phoneme | **0.312** | 0.260 | 0.196 | 0.289 | 0.310 | 0.294 |

## C.9  NOISE RATIO ANALYSIS

This experiment aims to assess the robustness of the CoInception framework under various noise ratios within a given dataset. Additionally, it aims to demonstrate the partial enhancement of noise resilience achieved by CoInception, particularly through its focus on the high-frequency component. For comparison, we also verify this characteristic of TS2Vec Yue et al. (2022). For this experiment, we select forecasting as the representative task, using the ETTm1 dataset. By introducing random Gaussian noises with a mean equal to $x\%$ of the input series's mean attitude in pretraining stage, the goal is for two models to learn efficient representations even with the presence of noise. The current experiment sets $x$ to be 10, 20, 30, 40, and 50, as going beyond these values would result in the noise outweighing the underlying series, making it impractical to be considered as noise. We also report the results without noise (noted as $x = 0$) for complete reference. It is understandable that the model performance deteriorates when the noise level increases.

| Dataset | TS2Vec | TS2Vec* | T-Loss | T-Loss* | CoInception | CoInception* |
|---|---|---|---|---|---|---|
| Plane | **1.000** | 0.981 | 0.981 | 0.990 | **1.000** | **1.000** |
| ProximalPhalanxOutlineCorrect | 0.887 | 0.876 | 0.869 | 0.859 | **0.911** | 0.893 |
| ProximalPhalanxOutlineAgeGroup | 0.834 | 0.844 | 0.839 | **0.854** | 0.849 | 0.844 |
| ProximalPhalanxTW | **0.824** | 0.805 | 0.785 | **0.824** | **0.824** | 0.820 |
| RefrigerationDevices | 0.589 | 0.557 | 0.555 | 0.517 | 0.597 | **0.635** |
| ScreenType | 0.411 | 0.421 | 0.384 | 0.413 | 0.413 | **0.469** |
| ShapeletSim | **1.000** | **1.000** | 0.517 | 0.817 | 0.994 | **1.000** |
| ShapesAll | **0.902** | 0.877 | 0.837 | 0.875 | 0.898 | 0.863 |
| SmallKitchenAppliances | 0.731 | 0.747 | 0.731 | 0.715 | **0.792** | 0.717 |
| SonyAIBORobotSurface1 | 0.903 | 0.884 | 0.840 | 0.897 | **0.908** | 0.903 |
| SonyAIBORobotSurface2 | 0.871 | 0.872 | 0.832 | 0.934 | 0.939 | **0.940** |
| StarLightCurves | 0.969 | 0.967 | 0.968 | 0.965 | 0.971 | **0.974** |
| Strawberry | 0.962 | 0.962 | 0.946 | 0.946 | **0.970** | **0.970** |
| SwedishLeaf | 0.941 | 0.931 | 0.925 | 0.931 | 0.950 | **0.957** |
| Symbols | **0.976** | 0.973 | 0.945 | 0.965 | 0.970 | 0.961 |
| SyntheticControl | **0.997** | **0.997** | 0.977 | 0.983 | **0.997** | 0.990 |
| ToeSegmentation1 | 0.917 | 0.947 | 0.899 | **0.952** | 0.943 | 0.947 |
| ToeSegmentation2 | 0.892 | **0.946** | 0.900 | 0.885 | 0.908 | 0.900 |
| Trace | **1.000** | **1.000** | **1.000** | **1.000** | **1.000** | **1.000** |
| TwoLeadECG | 0.986 | **0.999** | 0.993 | 0.997 | 0.998 | **0.999** |
| TwoPatterns | **1.000** | 0.999 | 0.992 | **1.000** | **1.000** | **1.000** |
| UWaveGestureLibraryX | 0.795 | 0.818 | 0.784 | 0.811 | 0.817 | **0.820** |
| UWaveGestureLibraryY | 0.719 | **0.739** | 0.697 | 0.735 | **0.739** | 0.738 |
| UWaveGestureLibraryZ | 0.770 | 0.757 | 0.729 | 0.759 | **0.771** | 0.754 |
| UWaveGestureLibraryAll | 0.930 | 0.918 | 0.865 | 0.941 | 0.937 | **0.956** |
| Wafer | 0.998 | 0.997 | 0.995 | 0.993 | **0.999** | 0.998 |
| Wine | 0.870 | 0.759 | 0.685 | 0.870 | **0.907** | **0.907** |
| WordSynonyms | 0.676 | 0.693 | 0.641 | **0.704** | 0.683 | 0.691 |
| Worms | 0.701 | **0.753** | 0.688 | 0.714 | 0.740 | 0.701 |
| WormsTwoClass | 0.805 | 0.688 | 0.753 | **0.818** | **0.818** | 0.779 |
| Yoga | **0.887** | 0.855 | 0.828 | 0.878 | 0.882 | 0.854 |
| Avg. (first 85 datasets) | 0.829 | 0.824 | 0.786 | 0.821 | **0.841** | **0.829** |

Figure 14 and the quantitative results in Table 13 summarize our findings with this experiment. In general, while both methods illustrate the decrease in performance upon the introduction of noise, CoInception still consistently outperforms TS2Vec, suggested by the performance decrease (in percentage) of TS2Vec compared with CoInception in Table 13. This results attributes to our strategy to ensure noise-resilience toward high-frequency noisy components.

Figure 13: CoInception and TS2Vec performance with different noise ratio in ETTm1 dataset.

| Noise Ratio | | CoInception | TS2Vec |
|---|---|---|---|
| 0% | MSE | 0.061 | 0.069 (-11.59%) |
| | MAE | 0.173 | 0.186 (-6.98%) |
| 10% | MSE | 0.17 | 0.203 (-16.25%) |
| | MAE | 0.332 | 0.364 (-8.79%) |
| 20% | MSE | 0.175 | 0.209 (-4.79%) |
| | MAE | 0.336 | 0.369 (-8.94%) |
| 30% | MSE | 0.177 | 0.21 (-15.71%) |
| | MAE | 0.339 | 0.37 (-8.27%) |
| 40% | MSE | 0.18 | 0.211 (-14.69%) |
| | MAE | 0.342 | 0.371 (-7.81%) |
| 50% | MSE | 0.181 | 0.213 (-15.02%) |
| | MAE | 0.343 | 0.371 (-7.54%) |

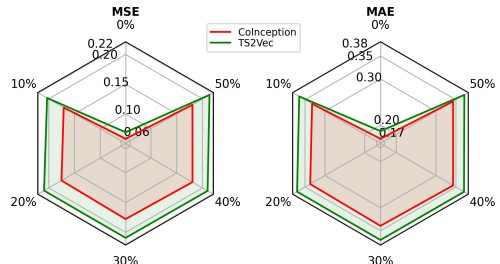

Figure 14: Assessing CoInception and TS2Vec performance with exposure to different noise ratio in ETTm1 dataset.

## D    ADDITIONAL DICUSSIONS REGARDING COINCEPTION

This section is dedicated to discussing certain limitations and potential drawbacks of the CoInception framework. These insights aim to assist readers in determining suitable applications for CoInception.

About the sampling strategy based on DWT, we implicitly limit the noise being targeted in this study to be high-frequency. By removing the high-frequency components, the filter helps to smooth out the signal and eliminate rapid fluctuations caused by noise, while better revealing the underlying trends or slow-varying patterns in the time series. However, this strategy does not necessarily create an ideal noise-free signal of the series. In the circumstance where the dataset is either completely free of noise

or inherently possesses noise predominantly in the low-frequency spectrum (such as a drifting effect Smith et al. (1999); He et al. (2019)), our proposed strategy might not offer significant benefits in managing those noisy signals.

About the encoder architecture, while the current design aligns with our main criteria of reaching both efficiency and effectiveness, it comes with a potential trade-off. With the use of Inception idea to automate the choice of scaling dilation factors, the problem of optimizing the number of layers used remains to be answered. This problem is also related to efficiency-effectiveness trade-off, hence it still needs extra effort to determine the number of layers used in the architecture. While we currently limit and fix our framework with 3 layers, we make no claim about the optimal number of layers to be used, but should be fine-tuned instead depending on the tasks or datasets specifically. We would consider this factor for a future study.

