# OpenReview forum: "Enriching Time Series Representation: Integrating a Noise-Resilient Sampling Strategy with an Efficient Encoder Architecture"
_ICLR.cc/2024/Conference — ICLR 2024 Conference Withdrawn Submission_

### Official Review · Reviewer_sJ4A · 2023-10-31

**Soundness:** 3 good
**Presentation:** 3 good
**Contribution:** 3 good
**Rating:** 6
**Confidence:** 4

**Summary:**

This paper presents a self-supervised learning framework that explicitly considers noise and ensures consistent representation given noisy input data. The proposed architecture is also lightweight and efficient. Experiments across three tasks (forecasting, classification, anomaly detection) verify the effectiveness of the proposed method.

**Strengths:**

1. The proposed representation learning framework explicitly takes noise into consideration, and enables consistent representation despite noise in the original data.
2. The proposed model architecture with Inception blocks is lightweight and efficient.
3. Extensive experiments on three time-series related downstream datasets demonstrate promising performance.

**Weaknesses:**

1. The direct forecasting baselines compared in the paper are no longer the state of the art. Including more recent baselines like PatchTST and DLinear could better showcase the effectiveness of the proposed method in comparative analysis.
2. The diversity of forecasting datasets seems limited, focusing only on the electricity domain.
3. How to quantify the performance improvement with respect to the noise levels in the original datasets? For example, do higher noise levels in the original datasets correspond to larger performance improvement after adopting the proposed noise-resilient approach?
4. The choice of Discrete Wavelet Transform (DWT) for denoising, as opposed to other frequency-domain denoising techniques like high-frequency component filtering post-Fast Fourier Transform (FFT), needs further clarification.
5. The paper could benefit from clarifying any underlying assumptions about the noise characteristics, such as whether the noise needs to be additive rather than multiplicative in relation to the original time series.

**Questions:**

See Weaknesses above.

---

> ### Author Response · Authors · 2023-11-14
> **Response to Reviewer sJ4A**
>
> Thank you for dedicating your time to reviewing our paper and providing detailed, informative comments. Besides the responses, we have uploaded our revised manuscript and appendix, with all the revisions and additional details made based on your constructive feedback. Below are our responses to your specific questions:
>
> *1. Regarding forecasting baselines*
>
> Thank you for sharing the interesting studies, and we have completed the reading of these remarkably interesting works. We cite these references in our revised manuscript in the literature reviews section. We are working to add these methods into our experiments. However, due to the short rebuttal time and extensive experiments, we can not guarantee that such comparisons appear within the time period. Nevertheless, we will do our best to add the references to the experiments in the next version of our manuscript.
>
> *2. The diversity of forecasting datasets seems limited, focusing only on the electricity domain.*
>
> About the use of 4 datasets for forecasting task, there are several reasons for us to do so:
> - They are widely used to assess performance of time series frameworks when predicting long future sequences.
> - We incorporate these datasets in line with earlier research [1,2], and we follow the specific configurations outlined in those studies. This procedure is taken to minimize potential biases arising from varying experimental conditions, ensuring a fair comparison of our method with existing research.
> - Three of four datasets come from the same repository (ETT datasets), and one from UCI repository. These 2 repositories have the distinguished target variables: For ETT datasets, the target variable is Oil Temperature (OT), while the variable of Electricity dataset is electrical consumption in kWh. This diversity enhances the overall Forecasting section, rendering it more comprehensive and universally applicable.
>
> *3. How to quantify the performance improvement with respect to the noise levels in the original datasets? For example, do higher noise levels in the original datasets correspond to larger performance improvement after adopting the proposed noise-resilient approach?*
>
> We thank the Reviewer for this insightful comment. Following this suggestion, we plan to further design and conduct an experiment investigating the relationship between noise levels and models’ performance. We are working to add experimental result in the next version of our manuscript.  However, due to the short rebuttal time and extensive experiments, we can not guarantee that such comparisons appear within the time period.
>
> *4. Clarification on the choice of Discrete Wavelet Transform (DWT)*
>
> Our initial preference for the Discrete Wavelet Transform (DWT)-based low-pass filter is grounded in several considerations outlined below:
> - DWT, as opposed to another commonly used method in frequency analysis, FFT, offers a distinct advantage that aligns with our objectives in the low-pass filter. While FFT provides a decomposition without temporal information, DWT deconstructs series into multiple frequency sub-bands with reduced time resolutions. Opting for DWT grants us more flexibility in determining which frequency bands to diminish at specific time ranges, allowing targeted reduction without completely eliminating entire frequency spectra from the series of interest.
> - This filter operates as an independent module and requires no training parameters, making it decoupled from other modules in the CoInception framework. This modular design opens space for potential replacements that may be better suited to specific datasets or tasks.
>
> It's crucial to underscore that our primary contribution lies not in the low-pass filter itself but in the objective of generating noise-resilient representations. Consequently, our current selection of the DWT low-pass filter may not be universally optimal for all datasets or tasks. For particular datasets, we believe spending extra effort to carefully choose the suitable low-pass filter could enhance the CoInception pipeline’s performance even further.

---

> > ### Author Response · Authors · 2023-11-14
> > **Response to Reviewer sJ4A (cont)**
> >
> > *5. Assumptions about the noise characteristics*
> >
> > We thank you for this suggestion. We would like to clarify that the reference to additive noise in our manuscript serves a demonstrative purpose only, just like [3,4] only showcasing this type of noise, to indicate the presence of noise components in time series signals. It's important to note that our mention of additive noise is not indicative of any assumption regarding its prevalence compared to other types of noise.
> > Besides, other than this mentioning, the additive-ness of noisy signals is not related to the design as well as the functioning of any modules in the CoInception framework. We have amended the corresponding paragraph mentioning additive noise, and make better clarification to avoid any further misunderstanding.
> >
> > In summary, we hope our responses clarify all your questions. We express our gratitude once again for your positive evaluations of our paper. Upon the resolution of all your concerns, we sincerely hope that you may consider increasing your score and offering your support.
> >
> >
> >
> > [1] Zhou, Haoyi, et al. "Informer: Beyond efficient transformer for long sequence time-series forecasting." Proceedings of the AAAI conference on artificial intelligence. Vol. 35. No. 12. 2021.
> >
> > [2] Yue, Zhihan, et al. "Ts2vec: Towards universal representation of time series." Proceedings of the AAAI Conference on Artificial Intelligence. Vol. 36. No. 8. 2022.
> >
> > [3] "Digital Signal Processing: Principles, Algorithms, and Applications" by John G. Proakis and Dimitris G. Manolakis
> >
> > [4] "Introduction to Signal Processing" by Sophocles J. Orfanidis

---

> > > ### Author Response · Authors · 2023-11-19
> > > **Response to Reviewer sJ4A (cont)**
> > >
> > > Dear Reviewer sJ4A,
> > >
> > > We have updated our manuscript, with the additional analysis investigating our framework performance toward different levels of noise.
> > >
> > > To be more specific, we choose forecasting, using the ETTm1 dataset, as the representative task. In the pretraining stage, we introduce random Gaussian noises with a mean equal to x% of the input series's mean attitude (here x ranging from 0 to 50). We further put our method into comparison with the current state-of-the-art [5]. In general, while both methods illustrate the decrease in performance upon the introduction of noise, CoInception still consistently outperforms the baseline, which attributes to our strategy to ensure noise-resilience toward high-frequency noisy components.
> > > Here, we present a table summarizing the results of this experiment, where the percentages indicate the comparisons between the performance of TS2Vec and CoInception, within the same noise levels.
> > >
> > > | Noise Ratio |     |              CoInception              |     TS2Vec      |
> > > |:------------|:----|:-------------------------------------:|:---------------:|
> > > | 0%          | MSE | **0.061** | 0.069 (-11.59%) |
> > > |             | MAE | **0.173** | 0.186 (-6.98%)  |
> > > | 10%         | MSE | **0.17**  | 0.203 (-16.25%) |
> > > |             | MAE | **0.332** | 0.364 (-8.79%)  |
> > > | 20%         | MSE | **0.175** | 0.209 (-4.79%)  |
> > > |             | MAE | **0.336** | 0.369 (-8.94%)  |
> > > | 30%         | MSE | **0.177** | 0.21 (-15.71%)  |
> > > |             | MAE | **0.339** |  0.37 (-8.27%)  |
> > > | 40%         | MSE | **0.18**  | 0.211 (-14.69%) |
> > > |             | MAE | **0.342** | 0.371 (-7.81%)  |
> > > | 50%         | MSE | **0.181** | 0.213 (-15.02%) |
> > > |             | MAE | **0.343** | 0.371 (-7.54%)  |
> > >
> > >
> > > We hope that this experiment provides additional clarification regarding your raised question. We are also open to answer any additional questions you may have. Once all your concerns are resolved, we sincerely hope you might consider raising your score and support our work.
> > >
> > > Best regards,
> > >
> > > Authors.
> > >
> > >
> > > [5] Zhihan Yue, Yujing Wang, Juanyong Duan, Tianmeng Yang, Congrui Huang, Yunhai Tong, and Bixiong Xu. Ts2vec: Towards universal representation of time series. In Proceedings of the AAAI Conference on Artificial Intelligence, volume 36, pp. 8980–8987, 2022.

---

### Official Review · Reviewer_TahQ · 2023-11-01

**Soundness:** 4 excellent
**Presentation:** 4 excellent
**Contribution:** 3 good
**Rating:** 6
**Confidence:** 4

**Summary:**

Ths paper proposes three improvements to learn a better representation of time series data, especially under noise situation. The three parts  are a novel sampling strategy, an enhanced dilated convolutional encoder, and a new triplet loss. The experiments on different domains of time series dataset, show consistent improvement over the state-of-the-art TS2VEC model.

**Strengths:**

1. The authors provide ablation study on how effective of each proposed components, and show all of them contributes to the final improvement.
2. The testing dataset is ocmprehensive, including popular UCR, UEA, Yahoo, etc. The results are evaluated on most of the subset in each repository.
3. The domain of experiments is also comprehensive, including forecasting, classification, abnormaly detection, etc. A consistent improvement on all the domain and all the dataset make the contribution very solid.

**Weaknesses:**

1. It might be better to see an additional ablation study on the encoder architecture, as the authors also proposes 3 changes on the original encoder.
2. The noise reduction is demonstrated in Figure 2, but it is hard to tell what to expect from the figure. I would suggest using some statistics to show the closeness between the embeddings of original time series and the perturbed ones.

**Questions:**

The noise reduction effect is mainly attributes to which of the three proposed compoenents?

---

> ### Author Response · Authors · 2023-11-14
> **Response to Reviewer TahQ**
>
> Thank you for your time and the positive evaluations of our paper. We also express gratitude for your constructive comments and suggestions. Besides the responses, we have uploaded our revised manuscript and appendix, with all the revisions and additional details made based on your constructive feedback. We hope that our responses below sufficiently address all of your questions:
>
> *1. An additional ablation study on the encoder architecture*
>
> We thank you for these great constructive comments. We find this experiment can be directly addressed, given that it is related to our framework and worth investigating. We plan to update these results together with our revised manuscript in the next few days.
>
> *2. The noise reduction is demonstrated in Figure 2, but it is hard to tell what to expect from the figure.*
>
> Thank you for pointing out this. We have made our motivation clearer with Figure 2, including quantitative correlation calculation measured by cosine similarity to show the proximity between the representations of original time series and the perturbed ones.
>
> *3. The noise reduction effect is mainly attributes to which of the three proposed compoenents?*
>
> The initiation of the noise reduction effect occurs during the sampling stage for training data samples, where segments of both raw and disturbed series are selected. Subsequently, the actual realization of this effect is guided by our alignment loss and stabilized by the Triplet Loss regularization term through the training process.
>
> Nevertheless, while the CoInception Encoder does not directly support the noise reduction effect, its contribution is also crucial in the whole pipeline. It balances efficiency and effectiveness, contributing significantly to the enhanced performance of CoInception, as demonstrated in Table 4 of the ablation experiment. Moreover, owing to the independent nature of these two modules, future studies might be benefited from different aspects of the CoInception pipeline, such as the reutilization of the CoInception architecture, without necessitating modifications or assumptions about other modules.
>
> Once again, thank you for your positive evaluations of our paper. Please leave us with other responses if you have any further concerns. When all your concerns are resolved, we sincerely hope that you could consider increasing your score and support us.

---

> > ### Author Response · Authors · 2023-11-19
> > **Response to Reviewer TahQ (cont)**
> >
> > Dear Reviewer TahQ,
> >
> > We have updated our manuscript, including the additional experiment investigating the effect of different ablations on CoInception encoders’s performance.
> >
> > Specifically, three changes were made: **(2a)** Replaced the Aggregator with simple concatenation and added a Bottleneck layer based on [1]; **(2b)** Replaced Dilated Convolutions with regular 1D Convolution layers; **(2c)** Removed skip connections between Basic Units of different layers. Table below summarizes the results, consistently showing a decline in overall performance when any suggested alteration is excluded or replaced, emphasizing the positive impact on CoInception framework robustness.
> >
> > |Version|Classification| |Forecasting (ETTm1 - avg)| |Abnormally Detection| | |
> > |:---:|:---:|:---:|:---:|:---:|:---:|:---:|:---:|
> > | |Acc.|AUC.|MSE|MAE|F1|P.|R.|
> > |W/o Aggregator|0.671|0.734|0.068|0.186|0.648|0.626|0.671|
> > |W/o Dilated Conv|0.571|0.635|0.064|0.178|0.647|0.639|0.656|
> > |W/o Skip Connection|0.654|0.719|**0.06**|**0.172**|0.653|0.617|0.694|
> > |Full version|**0.705**|**0.774**|0.061|0.173|**0.769**|**0.790**|**0.748**|
> >
> > We hope this experiment further clarifies your raised question. Also, we are happy to answer any additional questions from you. Upon the resolution of all your concerns, we sincerely hope that you may consider increasing your score to support our work.
> >
> >
> > Best regards,
> >
> > Authors.
> >
> > [1] Haixu Wu, Tengge Hu, Yong Liu, Hang Zhou, Jianmin Wang, and Mingsheng Long. Timesnet: Temporal 2d-variation modeling for general time series analysis. arXiv preprint arXiv:2210.02186, 2022.

---

> > > ### Comment · Reviewer_TahQ · 2023-11-23
> > > **Thank you for your response**
> > >
> > > I acknowledge that I have read authors' responses and I appreciate the additional ablation studies by the authors. I will stick to my positive score.

---

> > > > ### Author Response · Authors · 2023-11-23
> > > > **Thank you for the positive review**
> > > >
> > > > Dear Reviewer TahQ,
> > > >
> > > > We thank you and appreciate your positive evaluation of our work. We hope you support the publication of this work as this motivates us to further pursue this research direction.
> > > >
> > > > Again, thank you for your time and considerations.
> > > >
> > > > Best regards,
> > > >
> > > > Authors.

---

### Official Review · Reviewer_CP3P · 2023-11-01

**Soundness:** 2 fair
**Presentation:** 2 fair
**Contribution:** 2 fair
**Rating:** 5
**Confidence:** 4

**Summary:**

This paper presents a new time-series representation learning strategy with contrastive learning, which highlights the importance of handling noise. Specifically, they propose the noise-resilient sampling strategy to find the positive and negative pairs where different views of the data samples are created from the spectral domain instead of the temporal domain. They present a new time-series encoder that leverages Inception and Dilation to achieve efficiency and wide receptive field at the same time.

**Strengths:**

1. Their proposed new encoder architecture is both lightweight and able to look at a wide receptive field.
2. They’ve conducted extensive experiments to show the advantages of their proposed methods in time-series representation learning in the tasks of forecasting, classification, and anomaly detection.

**Weaknesses:**

1. One of the key assumptions that the authors make is that noise is usually in high frequency, citing two reference papers (Lanting et al., 2011 and Oohashi et al., 2000). But I highly doubt whether this is the case in general. The 1st reference paper (Lanting et al., 2011) focuses on a very niche case (Macroscopic Resonant Tunneling), while the key insight of the 2nd one (Oohashi et al., 2000) is not even related based on my glimpse at the abstract of that paper. Thus, I would doubt the usefulness of the proposed method.
2. Writing and English need to be improved. I’ll give 3 examples but there are more scattering around the entire paper. (1) for Figure 2, it is very confusing which part I should be looking at when reading the corresponding text in Section 2.2. (2) On page 4, it is confusing what level $j$ refers to without further explanation (I could sort of infer it means the $j$-th dimension of $\mathbf{x}$). (3) The long paragraph on Page 5 should be better structured and many sentences in it are informal and unprofessional. I would suggest the authors read through the paper before submitting it.

**Questions:**

I have included some of the questions in the part of Weakness. Other than that:
- What are the characteristics of datasets that you expect your proposed method to work better on? For example, do they tend to have noise in distinct frequencies or require longer sequence dependence?

---

> ### Author Response · Authors · 2023-11-14
> **Response to Reviewer CP3P**
>
> We appreciate your time and effort spent on reviewing our manuscript. We are glad to receive your acknowledgment of our encoder design and our efforts in conducting experiments. Besides the responses, we have uploaded our revised manuscript and appendix, with all the revisions and additional details made based on your constructive feedback. We address and clarify these concerns point-by-point below:
>
> *1. Regarding high frequency noise*
>
> Thank you for this insightful comment. We are sorry that our sentences gave a false impression that we assume high frequency in noise. We agree with you about the two references and we remove them since they potentially lead to a misunderstanding of our intended idea. We hope to discuss this in detail and clarify our work better with you.
>
> While we focus on addressing high-frequency noise, we do not necessarily assume that noise is inherently high frequency. However, we have motivations to consider this setting, which is the observation that high-frequency noisy components often correspond to short-term variations or bursts that may not significantly contribute to the overall structure of the time series. The contextual information derived from these short-term fluctuations can be misleading and may not accurately represent the underlying series of interest. Therefore, our objective is to make the CoInception framework insensitive to these high-frequency noise. By doing so, we aim to ensure that the output representations are more robust to variations and disturbances in the input. In addition, this approach aligns well with various studies [1,2,3,4] emphasizing the utilization of disentangled components of raw series, such as seasonality or trends, which exhibit persistence in a long-term context.
>
> For the above reasons, we argue that the study of this setting is useful and beneficial in practice, and this fact is proven by our extensive experiments. In the revision, we clearly explain why we are specifically addressing high-frequency noise and update our references accordingly.
>
> *2. Regarding our writing*
>
> Thank you for this straight and constructive feedback. As mentioned, we have refined our writing in this revision (taking into account all three examples you mentioned), carefully with our choices of words to avoid any potential misunderstandings and concerns. If you see any other examples, please let us know and we will work with you promptly to make sure our manuscript is clear and concise.
>
> *3. What are the characteristics of datasets that you expect your proposed method to work better on? For example, do they tend to have noise in distinct frequencies or require longer sequence dependence?*
>
> We thank you for an insightful question on the further analysis of our methods. While our primary focus is on making the CoInception framework robust to high-frequency noise, it's important to note that we don't presume superior performance on specific datasets. This intentional lack of expectation ensures the versatility of our design, allowing it to be applicable across a broad spectrum of datasets and tasks.
> Specifically, when defining the 'high-frequency' spectrum in our framework, we don't restrict it to a fixed range of frequency bands. Instead, it is calculated based on the characteristics of the datasets to which the CoInception framework is applied. This adaptability feature allows the CoInception network to flexibly adjust to various datasets and tasks.
> It's worth mentioning another potential concern, constructively mentioned by Reviewer CkvB: the risk of over-smoothing effect in CoInception when high-frequency components are informative for representing the original series. We efficiently address it with our Triplet loss regularization term by prioritizing alignment between two raw overlapped series. Further details on this matter can be found in our original response to the first question posed by Reviewer CkvB.
>
> With these responses, we hope that we have addressed all of your current concerns. Should you need any other clarification, please leave a comment and we are happy to further discuss with you. Given your appreciation for the CoInception encoder design and our comprehensive experiments, we sincerely hope that once all your concerns are resolved, you may reconsider the paper, potentially increasing your score and supporting our approach. Thank you for your time and consideration.

---

> > ### Author Response · Authors · 2023-11-14
> > **Response to Reviewer CP3P (cont)**
> >
> > [1] Wang, Zhiyuan, et al. "Learning latent seasonal-trend representations for time series forecasting." Advances in Neural Information Processing Systems 35 (2022): 38775-38787.
> >
> > [2] Woo, Gerald, et al. "Etsformer: Exponential smoothing transformers for time-series forecasting." arXiv preprint arXiv:2202.01381 (2022).
> >
> > [3] Woo, Gerald, et al. "CoST: Contrastive learning of disentangled seasonal-trend representations for time series forecasting." arXiv preprint arXiv:2202.01575 (2022).
> >
> > [4] Huang, Siyuan, et al. "CrossWaveNet: A dual-channel network with deep cross-decomposition for Long-term Time Series Forecasting." Expert Systems with Applications 238 (2024): 121642.

---

> > > ### Author Response · Authors · 2023-11-17
> > > **Response to Reviewer CP3P (cont)**
> > >
> > > Dear Reviewer CP3P,
> > >
> > > We wanted to inform you that we've made the suggested changes to the PDF. We hope our responses address all your inquiries.
> > >
> > > In case you still have additional concerns, we would be more than happy to clarify. Your prompt feedback within this discussion period would be greatly appreciated. If you find that all your questions have been satisfactorily resolved, we kindly request you to consider adjusting your score to support our research efforts.
> > >
> > > Regards,
> > >
> > > Authors

---

> > > > ### Comment · Reviewer_CP3P · 2023-11-19
> > > >
> > > > Dear authors,
> > > >
> > > > Thanks for your response and the revisions made to the manuscript.
> > > >
> > > > I can see you removed the previous two references for "the noise-like elements, which appear in high frequency spectrum of the original series". I understand that your paper specifically addresses the challenge posed by noise induced by high-frequency components, and this seems to be a general issue in many time series data. But without legitimate citations or empirical evidence regarding high-frequency noise, I still cannot fully understand the source of performance gain of your proposed method.
> > > >
> > > > The writing of the revised version seems to be improved.
> > > >
> > > > Regarding my inquiry about the characteristics of datasets where your proposed method might work well, I was hoping to gain more insights into the circumstances under which your approach outperforms others. Acknowledging that there is no universally "best" model applicable to all datasets, discussing scenarios or types of data where your method is particularly advantageous would be valuable. This information would guide potential users in making informed decisions about when to employ your method.
> > > >
> > > > I have updated my score.
> > > >
> > > > Best,

---

> > > > > ### Author Response · Authors · 2023-11-19
> > > > > **Response to Reviewer CP3P (cont)**
> > > > >
> > > > > Dear Reviewer CP3P,
> > > > >
> > > > > Thank you for boosting up your score and further informing us with your remaining concerns. We would like to provide you with additional clarifications regarding these two points, along with corresponding modifications in our manuscript:
> > > > >
> > > > > *About legitimate citations or empirical evidence regarding high-frequency noise*
> > > > >
> > > > > Thank you for your question. We would like to discuss more with you about this and provide some additional references for works targeting high-frequency noise as their major or partial objectives: [5,6,7,8,9].
> > > > >
> > > > > While these applications inspire our method, we do not attach CoInception framework to just these uses. Our approach extends its benefits to datasets with noise signals across a wide frequency spectrum. By filtering out non-representative high-frequency information during learning, our method aims to ensure that the output representations remain at least invariant to the high-frequency components of those noisy signals. This partial improvement contributes to enhancing the noise-resilience of learned embeddings in those applications.
> > > > >
> > > > > To reinforce our findings regarding the partial enhancement of CoInception in handling noisy signals across a broad frequency spectrum, we conducted an additional experiment. This experiment involved introducing Gaussian white noises, which cover a wide frequency band, into the ETTm1 dataset.
> > > > > Our experimental results suggested that CoInception has the ability to partially resist the impact of noise, which further attributes to its superior performance compared to a state-of-the-art method. To be specific, while adding noise may deteriorate the performance of our methods to some extent, it still performs consistently better than other methods within the same level of noise. Below, we provide the Table summarizing results of this experiment, where the percentages indicate the comparisons between the performance of TS2Vec and CoInception, within the same noise levels.
> > > > >
> > > > > | Noise Ratio |     |              CoInception              |     TS2Vec      |
> > > > > |:------------|:----|:-------------------------------------:|:---------------:|
> > > > > | 0%          | MSE | **0.061** | 0.069 (-11.59%) |
> > > > > |             | MAE | **0.173** | 0.186 (-6.98%)  |
> > > > > | 10%         | MSE | **0.17**  | 0.203 (-16.25%) |
> > > > > |             | MAE | **0.332** | 0.364 (-8.79%)  |
> > > > > | 20%         | MSE | **0.175** | 0.209 (-4.79%)  |
> > > > > |             | MAE | **0.336** | 0.369 (-8.94%)  |
> > > > > | 30%         | MSE | **0.177** | 0.21 (-15.71%)  |
> > > > > |             | MAE | **0.339** |  0.37 (-8.27%)  |
> > > > > | 40%         | MSE | **0.18**  | 0.211 (-14.69%) |
> > > > > |             | MAE | **0.342** | 0.371 (-7.81%)  |
> > > > > | 50%         | MSE | **0.181** | 0.213 (-15.02%) |
> > > > > |             | MAE | **0.343** | 0.371 (-7.54%)  |
> > > > >
> > > > > This additional experiment is described in Section C.9, revised Appendix.
> > > > >
> > > > > In addition, our current objective is highly-correlated with many lines of works studying time series representation learning (e.g studies using disentangled components like trend or seasonal - as mentioned in the revised manuscript or in our previous response; studies applying denoising techniques [10,11,12]).
> > > > >
> > > > > *About source of performance gain of your proposed method*
> > > > >
> > > > > We emphasize that the root of our performance gain come from many factors:
> > > > > - CoInception is invariant to high-frequency noise-like components encompassed in all kind of data
> > > > > - Its encoder effectively strike the balance between efficiency and effectiveness
> > > > > - The triplet loss mitigate the effect of over-smoothing effect caused by excessive filtering of high-frequency components
> > > > >
> > > > > *About additional insights for the circumstances under which CoInception outperforms others*
> > > > >
> > > > > Thank you for your question. It is true that there is no universally "best" model applicable to all datasets. One of the scenarios where our current training method may not fully demonstrate its effectiveness is when the dataset is either completely free of noise or inherently possesses noise predominantly in the low-frequency spectrum (such as a drifting effect [13,14]), our proposed strategy might not offer significant benefits in managing those noisy signals. We additionally provide this insight in the Additional Discussion section of the revised manuscript.
> > > > >
> > > > > Again, we are open to answer any additional questions you may have. Once all your concerns are resolved, we sincerely hope you might consider raising your score and support our work.
> > > > >
> > > > > Best regards,
> > > > >
> > > > > Authors.

---

> > > > > > ### Author Response · Authors · 2023-11-19
> > > > > > **Response to Reviewer CP3P (cont)**
> > > > > >
> > > > > > [5] Lecocq, T., Hicks, S. P., Van Noten, K., Van Wijk, K., Koelemeijer, P., De Plaen, R. S., ... & Xiao, H. (2020). Global quieting of high-frequency seismic noise due to COVID-19 pandemic lockdown measures. Science, 369(6509), 1338-1343.
> > > > > >
> > > > > > [6] Jin, Z., Dong, A., Shu, M., & Wang, Y. (2019). Sparse ECG denoising with generalized minimax concave penalty. Sensors, 19(7), 1718.
> > > > > >
> > > > > > [7] Asgari, S., & Mehrnia, A. (2017). A novel low-complexity digital filter design for wearable ECG devices. PloS one, 12(4), e0175139.
> > > > > >
> > > > > > [8] Song, D., Baek, A. M. C., & Kim, N. (2021). Forecasting stock market indices using padding-based fourier transform denoising and time series deep learning models. IEEE Access, 9, 83786-83796.
> > > > > >
> > > > > > [9] Li, Y., Han, L., & Liu, X. (2023). Accuracy Enhancement and Feature Extraction for GNSS Daily Time Series Using Adaptive CEEMD-Multi-PCA-Based Filter. Remote Sensing, 15(7), 1902.
> > > > > >
> > > > > > [10] Rasul, K., Seward, C., Schuster, I., & Vollgraf, R. (2021, July). Autoregressive denoising diffusion models for multivariate probabilistic time series forecasting. In International Conference on Machine Learning (pp. 8857-8868). PMLR.
> > > > > >
> > > > > > [11] Morvan, M., Nikolaou, N., Yip, K. H., & Waldmann, I. (2022). Don't Pay Attention to the Noise: Learning Self-supervised Representations of Light Curves with a Denoising Time Series Transformer. arXiv preprint arXiv:2207.02777.
> > > > > >
> > > > > > [12] Dudek, G., Pełka, P., & Smyl, S. (2021). A hybrid residual dilated LSTM and exponential smoothing model for midterm electric load forecasting. IEEE Transactions on Neural Networks and Learning Systems, 33(7), 2879-2891.
> > > > > >
> > > > > > [13] Smith, A. M., Lewis, B. K., Ruttimann, U. E., Frank, Q. Y., Sinnwell, T. M., Yang, Y., ... & Frank, J. A. (1999). Investigation of low frequency drift in fMRI signal. Neuroimage, 9(5), 526-533.
> > > > > >
> > > > > > [14] He, X., Bos, M. S., Montillet, J. P., & Fernandes, R. M. S. (2019). Investigation of the noise properties at low frequencies in long GNSS time series. Journal of Geodesy, 93(9), 1271-1282.

---

> > > > > > > ### Author Response · Authors · 2023-11-21
> > > > > > > **Response to Reviewer CP3P (cont)**
> > > > > > >
> > > > > > > Dear Reviewer CP3P,
> > > > > > >
> > > > > > > We hope you've had a chance to review our latest response to your queries.
> > > > > > >
> > > > > > > As the discussion period is approaching its conclusion, we eagerly await your feedback. If there are any remaining concerns, we'd appreciate the opportunity for further discussion.
> > > > > > >
> > > > > > > Should our response address your questions satisfactorily, we kindly request your consideration in adjusting your score to support our work.
> > > > > > >
> > > > > > > Thank you for investing your time in reviewing and discussing our submission.
> > > > > > >
> > > > > > > Best regards,
> > > > > > >
> > > > > > > Authors

---

> > > > > > > > ### Author Response · Authors · 2023-11-22
> > > > > > > > **Response to Reviewer CP3P (cont)**
> > > > > > > >
> > > > > > > > Dear Reviewer CP3P,
> > > > > > > >
> > > > > > > > As today marks the final day of the discussion period, we wanted to check in and see if any lingering concerns or questions remain following our last response.
> > > > > > > >
> > > > > > > > If there are any aspects you would like to clarify or if you have additional questions, we are willing to engage in further discussion before the conclusion of the review process.
> > > > > > > >
> > > > > > > > Thank you for your time and consideration.
> > > > > > > >
> > > > > > > > Best regards,
> > > > > > > >
> > > > > > > > Authors.

---

### Official Review · Reviewer_UsRW · 2023-11-03

**Soundness:** 2 fair
**Presentation:** 3 good
**Contribution:** 2 fair
**Rating:** 3
**Confidence:** 3

**Summary:**

The paper presents "CoInception," a framework for time series representation learning that is both noise-resilient and efficient. Recognizing the challenges in time series analysis such as the presence of noise and the need for lightweight yet robust encoders, the authors present a novel sampling strategy alongside an encoder architecture designed to enhance noise resilience and task performance. Their sampling strategy utilizes a spectral low-pass filter to generate noise-invariant representations, ensuring that key time series features are preserved while reducing the influence of noise. For encoding, they combine Inception blocks with dilated convolutions to capture long-range dependencies within a scalable network architecture that remains computationally efficient.
The proposed framework is validated through experiments, showing superior performance in forecasting, classification, and anomaly detection tasks compared to existing methods. CoInception achieves this with fewer parameters, highlighting its efficiency.

**Strengths:**

The paper certainly represents a significant amount of work by the authors, especially in the experimental section. The related work section is also extensive.

**Weaknesses:**

The paper does not do a good job positioning itself in the vast literature of timeseries modeling. It is not clear how the paper extends the state of the art in this area, and whether/how the proposed method is new compared to some fairly basic signal processing concepts, such as low-pass filtering to reduce noise.

**Questions:**

The paper does not do a good job positioning itself in the vast literature of timeseries modeling. Some key statements in the paper show this issue more clearly. For example:

"A common shortcoming emerges in existing methods: none explicitly address noise in time series data alongside an appropriate strategy for handling this unwanted signal. Unlike other data types, real-world time series often harbor substantial noise, severely impacting task accuracy"

This statement is hard to accept. Of course all or most prior work in this area considers that the time series will be noisy. This is one of the main reasons we use neural networks, as opposed to other models that are much less capable to deal with noisy data.

Then, there are some sentences that are hard to understand. For example:

"we devise a sampling strategy based on the insight that the noisy signal combined with the original series shouldn’t disrupt time series frameworks. In essence, frameworks should produce consistent representations given noise-free or raw series (noise-resiliency characteristics)."

What "Frameworks" are the authors talking about?

Additionally, there are some aspects of the model, or claims about the model, that are very basic but they are presented as an important technical contribution. For example:

"To achieve this, we shift our focus from the temporal domain to the spectral domain. Here, we employ a spectrum-based low-pass filter to create correlated yet distinct views of each input time series. These augmented views serve as positive samples of the raw series, effectively capturing the desired noise invariance. The advantages of this low-pass filter-based augmentation are twofold: (1) the filter preserves key characteristics such as trend and seasonality, ensuring deterministic and interpretable representations; (2) it eliminates noise-prone high-frequency components, improving noise resilience and enhancing downstream task performance by aligning the raw signal representation with the augmented view"

Of course there is nothing particularly new in the previous paragraph. Working in the spectral domain and performing LPF to remove high-frequency noise are elementary operations that are typically taught at the level of undergraduate signal processing courses.

Considering the previous weaknesses, it is hard to provide specific technical comments to the authors, given that (at least to this reviewer) it is not clear if the paper has something really new to contribute in the area of timeseries modeling.

---

> ### Author Response · Authors · 2023-11-14
> **Response to Reviewer UsRW**
>
> Thank you for dedicating your time and effort to reviewing our paper. We appreciate your understanding of the work amount we invested in this study. At the same time, we are sorry that you found our claims being vague in this version. Here, we aim to address and clarify these concerns in our subsequent responses to specific questions and points you made. Besides the responses, we have uploaded our revised manuscript and appendix, with all the revisions and additional details made based on your constructive feedback. If further clarification is required, please feel free to leave a comment, and we would be delighted to engage in a discussion with you.
>
> Regarding our position in time series modeling literature, we kindly direct you to our common response to all Reviewers, where we offer a concise summary of all the contributions presented in our work. Furthermore, we assure you that in the revised version, we will exercise our utmost care to mitigate any potential ambiguity or misunderstanding arising from the choice of words.
>
> *1. Regarding how to deal with noises in time series data*
>
> While we agree with you that the intentional use of Neural Networks is partially attributed to their ability to cope with noise, we want to emphasize that this effort may not be sufficient to effectively diminish the impact of noise, especially when our goal is to produce robust and resilient time series representations.
> We visually highlight this point through Figure 2 in the manuscript. Considering the visually negligible impact of noise on the original series in this figure, we expect the fundamental characteristics in the learned representation to remain intact. However, we acknowledge the inconsistent representations generated by an existing State of The Art framework when presented with raw and noise-added series. This experiment is one of our motivations for explicitly considering noise resilience as a crucial characteristic of learned representations in the design of the CoInception framework.
>
>
> *2. What "Frameworks" are the authors talking about?*
>
> In these specific sentences, our intention is to underscore that the presence of noise in raw series should not impact the representations learned for the true underlying series under investigation. The term "frameworks" encompasses generic models aimed at producing robust time series representations for various downstream tasks.
> To enhance clarity, we revise the sentences as follows, and will accordingly update them in the revised manuscript:
> "we devise a sampling strategy guided by the principle that the presence of noise in the series should not hinder the functionality of our framework. Ideally, it should generate consistent representations whether provided with noise-free or raw series, highlighting noise-resiliency characteristics"
>
> *3. Regarding the basic aspects of the model but are presented as technical contributions*
>
> We apologize if you have the impression that these basic aspects are the technical contributions of our paper. However, we think this is just a misunderstanding and we have revised our writing to make sure there would be no confusion. As you rightly noted, this Low Pass Filter is well-established and extensively studied over its long history. Instead, we highlight its advantageous applications in the context of time series representation learning, aligning well with our initial goal: ensuring noise resilience characteristics. Furthermore, our current implementation of the LPF in the sampling stage effectively isolates it from other modules within the CoInception framework. Thus, the LPF serves as a tool assisting in achieving the ultimate goal of learning noise resiliency, and it can be replaced by any filters or modules that function to reduce potential noise in the given raw series, while other CoInception modules remain intact.
>
> Finally, we hope our answers address all of your questions. Given your acknowledgment of the effort invested in this study, we hope that once all your concerns are resolved, you may reconsider the paper, potentially increasing your score and supporting our approach. Thank you for your time and consideration.

---

> > ### Author Response · Authors · 2023-11-17
> > **Response to Reviewer UsRW**
> >
> > Dear Reviewer UsRW,
> >
> > We would like to let you know that we have updated the pdf to your suggestions. We hope our responses answer all your questions!
> >
> > In case you need any remaining clarifications, we would be more than happy to reply. Please let us know your thoughts as soon as you can (within this discussion period). If your questions are all properly addressed, we really hope that you consider increasing your score to support our work.
> >
> > Regards,
> >
> > Authors

---

> > > ### Author Response · Authors · 2023-11-19
> > > **Response to Reviewer UsRW (cont)**
> > >
> > > Dear Reviewer UsRW,
> > >
> > > We hope you have some time to consider our pdf which included answers to your question.
> > > Also, we are eager to engage in further discussion to address any remaining concerns you may have.
> > > If your concerns have been addressed all, we kindly hope you could consider to increase the score for supporting us.
> > >
> > > Since the deadline is approaching, we kindly want to hear back from you.
> > >
> > > Best regards,
> > >
> > > Authors.

---

> > > > ### Author Response · Authors · 2023-11-21
> > > > **Request for Feedback from Reviewer UsRW**
> > > >
> > > > Dear Reviewer UsRW,
> > > >
> > > > With the conclusion of the discussion period drawing near, we eagerly await your feedback on our latest response and the revised manuscript.
> > > >
> > > > If any concerns persist, we would appreciate the opportunity for further discussion. Your reconsideration of scores in the event of satisfactory resolution would be highly appreciated.
> > > >
> > > > Thank you for dedicating your time to read and review our paper.
> > > >
> > > > Best regards,
> > > >
> > > > Authors.

---

> > > > > ### Comment · Reviewer_UsRW · 2023-11-22
> > > > > **My initial concerns remain.**
> > > > >
> > > > > Dear Authors,
> > > > >
> > > > > On one hand, I appreciate that you have put a large effort in trying to address the reviewers' concerns (including mine) by providing lengthy explanations, additional experiments, etc.
> > > > >
> > > > > My main concerns about the paper, namely that it does not make a significant contribution in the vast literature of representation learning for time series, remains as that cannot be addressed with some more explanations or few additional experiments.

---

> > > > > > ### Author Response · Authors · 2023-11-22
> > > > > > **Response to Reviewer UsRW**
> > > > > >
> > > > > > Dear Reviewer UsRW,
> > > > > >
> > > > > > **We thank you for your reply and the time you spent reviewing this paper.**
> > > > > >
> > > > > > However, we respectfully disagree with your impression that the contributions are not significant.
> > > > > >
> > > > > > We emphasize that our paper has the following contributions:
> > > > > > 1. Our CoInception architecture aims to strike a balance between efficiency and effectiveness. The application of Inception layers in our architecture addresses the problem of tuning dilation factors in stacked Dilated Convolutional architecture. Thus **this combination enhances the performance of the model while still being lightweight** (demonstrated by our comprehensive empirical results). **This architecture can be used in a plug-and-play manner to any existing or new pipelines, which is another benefit this work brings to future study.**
> > > > > >
> > > > > > 2. Regarding the presence of noise, we emphasize that we contribute **in the specific context of self-supervised contrastive learning time series, not the general representation learning**. Note that in the contrastive learning time series literature, there are not many works considering noise resiliency as they mostly consider contextual/ temporal or augmentation invariance. Therefore, our contribution is still new. The application of existing techniques like DWT in our work is not an important factor in our contribution. **We believe why and how our pipeline uses those techniques are the things that matter.** Firstly, our empirical observation indicates that prior methods do not demonstrate noise resilience (Figure 2 in our manuscript). Secondly, other than applying LPF in our sampling strategy, **we specifically design our triplet loss function to take into account the presence of noise and the over-smoothing effect of de-noising techniques.** The regularization term helps our framework automatically soften the requirement of alignment between raw and perturbed series’ representations. With this modular design, our method does not necessarily stick with DWT or any specific techniques for de-noising. **This point effectively means that our method can be better along with the development of better techniques in signal denoising.**
> > > > > >
> > > > > > Given the significance of our framework and impressive empirical results, it is unfair if you disregard our entire contribution just because we use a well-established technique.
> > > > > >
> > > > > > Again, we are happy to engage in further discussion if you have any other questions.
> > > > > >
> > > > > > Thank you.
> > > > > >
> > > > > > Best regards,
> > > > > >
> > > > > > Authors.

---

### Official Review · Reviewer_CkvB · 2023-11-04

**Soundness:** 3 good
**Presentation:** 3 good
**Contribution:** 3 good
**Rating:** 6
**Confidence:** 3

**Summary:**

The paper proposes a novel framework called CoInception for time series representation learning. It addresses the challenges of noise and lack of efficient encoder architectures in time series tasks. The framework utilizes a noise-resilient sampling strategy and an encoder architecture with dilated convolution within the Inception block. Experimental results show that CoInception outperforms state-of-the-art methods in forecasting, classification, and anomaly detection tasks. The authors investigates the existence and impacts of noise in time series representation learning and introduces a noise-resilient sampling strategy to learn consistent representations despite the noise.

**Strengths:**

Originality: The paper introduces a novel noise-resilient sampling strategy and an efficient encoder architecture, which are not explicitly considered in previous works. It investigates the existence and impacts of noise in time series representation learning, addressing a critical factor that affects the efficacy of time series tasks.

Quality: The authors provide empirical validation of the proposed CoInception framework and compares it with recent state-of-the-art methods, demonstrating consistent outperformance in forecasting, classification, and anomaly detection tasks. The experiments highlight the effectiveness of the framework in learning informative representations robust to various downstream tasks.

Clarity: The paper clearly presents the motivation, challenges, and contributions of the research, as well as the design principles of the framework. It also provides a comprehensive evaluation of the proposed method, highlighting the best results for better comparison.

Significance: The authors address the gap in existing works by exploring the potential of unsupervised representation learning in time series data. The proposed framework offers a solution for learning informative representations without the need for costly and difficult labeling, particularly in privacy-sensitive fields like healthcare and finance. The experimental results demonstrate the superiority of the proposed method over state-of-the-art approaches, indicating its significance in advancing time series analysis.

**Weaknesses:**

1.This paper appears to be making modifications on top of TS2Vec, incorporating a denoising module and enhancing the encoder, which limits the novelty of this paper.

2.The authors do not provide a detailed analysis of the limitations and potential drawbacks of the proposed noise-resilient sampling strategy and encoder architecture. This could hinder a deeper understanding of the trade-offs and potential challenges in implementing and applying the CoInception framework in real-world scenarios.

3.Lack of comparison with more recent and diverse state-of-the-art methods in time series representation learning, beyond the ones mentioned in the paper. This could limit the comprehensive evaluation of the proposed CoInception framework and its performance against a wider range of approaches.

**Questions:**

1. In real-world time series data, there may be some high-frequency components. As mentioned by the authors, the DWT method in this paper is capable of filtering out high-frequency noise. How can useful high-frequency data be preserved and what trade-offs are made in addressing this issue?

2. Could you provide a clearer explanation of the Inception Block in the Method section, including an explanation of the meanings of variables like $b$ and $h$ in Figure 3 (a), as well as the significance of the numbers in brackets? This would help in better understanding this paper.

3. The comparison with baselines could be more comprehensive. I believe it would be more convincing if the authors could compare their method to recent approaches from the past two years, such as TimesNet [1].

[1] Wu, Haixu, et al. "TimesNet: Temporal 2D-Variation Modeling for General Time Series Analysis." The Eleventh International Conference on Learning Representations. 2022.

---

> ### Author Response · Authors · 2023-11-14
> **Response to Reviewer CkvB**
>
> We appreciate your precious comments and thank you for your recognition of our work on multiple aspects. We hope that our responses below answer all of your questions. Besides the responses, we have uploaded our revised manuscript and appendix, with all the revisions and additional details made based on your constructive feedback. Should you need any other clarification, please leave a comment and we are happy to discuss with you.
>
> *1. Regarding the novelty of this paper.*
>
> We agree that there are some alignments of our proposal with the previous work - TS2Vec, and understand your concern. However, we want to make further clarifications that those alignments are not overlapping with our contributions.
> - First, our novel noise-resilience sampling strategy is driven by its distinguished motivations. Concurrently, we ensure context-invariant characteristics through a modified version of the technique introduced with TS2Vec. The rationale behind this modification is well-founded, addressing potential collapses in certain scenarios, with detailed explanations provided in our Appendix A.1.
> - Second, it is important to note that the stacked Dilated Convolutional network itself is not a new concept and has been employed in prior works (e.g. [1]). We leverage and build upon this existing framework, as it aligns with our stated objective of achieving both robustness and efficiency.
>
> *2. Detailed analysis of the limitations and potential drawbacks of the proposed approach*
>
> Thank you for this suggestion. We hope to discuss this with you. We do rely on some assumptions as well as make some considerations upon designing the CoInception framework, which might be its limitations and potentially lead to drawbacks in some scenarios, listed below:
> - Regarding the DWT-based sampling strategy, our focus on high-frequency noise implicitly confines the type of noise being targeted in this study. By removing the high-frequency components, the filter helps to smooth out the signal and eliminate rapid fluctuations caused by noise, while better revealing the underlying trends or slow-varying patterns in the time series. However, this strategy does not guarantee to create an ideal noise-free signal of the series.
> - Regarding the encoder architecture, our current design, while making a positive step toward balancing efficiency and effectiveness, only partially attains this objective. Despite the automation of scaling dilation factors with the Inception idea, the challenge of optimizing the number of layers remains unanswered. This issue is inherent in the efficiency-effectiveness trade-off, indicating that we have not fully realized our goal. Presently constrained to a 3-layer encoder, we make no claim about the optimal number of layers to be used, recognizing the need for fine-tuning based on specific tasks or datasets.
>
> Given these insights, we will certainly take these factors into account for future work. Due to space limitations, we plan to incorporate these reflections in a dedicated section titled "Discussion" within the Appendix of the modified manuscript.
>
>
> *3. Regarding comparison with other state-of-the-art methods*
>
> We thank you for your suggestion, and we cite more references in our revised manuscript in the literature reviews section. We are working to add these methods into our experiments. However, due to the short rebuttal time and extensive experiments, we can not guarantee that such comparisons appear within the time period. Nevertheless, we will do our best to add the empirical comparisons (including TimesNet) in the next version of our manuscript.
>
> *4. Regarding high-frequency noise and trade-offs in addressing this issue*
>
> We account for this scenario in the design of the CoInception framework and address it implicitly through the introduction of a Triplet loss regularization term, detailed in Section 2.3 of the revised manuscript. Specifically, if the vital information of the original series is predominantly in the high-frequency spectrum, solely aligning raw samples' representations with disturbed ones could result in substantial information loss. The Triplet loss regularization term mitigates this effect by giving more priority to the alignment between two raw overlapped samples, thus alleviating the loss of essential information. Empirical results presented in Table 4 highlight the detrimental impact of ablating this Triplet term.

---

> > ### Author Response · Authors · 2023-11-14
> > **Response to Reviewer CkvB (cont)**
> >
> > *5. A clearer explanation of the Inception Block in the Method section*
> >
> > In Figure 3(a), the numbers within brackets represent the 'base kernel size' for three Basic Units within a CoInception layer. These numbers are then employed to calculate the corresponding dilation factors and accumulated receptive fields for those Units. Intentionally chosen, these three numbers cover different scales of dilation factors and accumulated receptive fields. Units with smaller base kernel sizes are designed to focus more on capturing short-term/local temporal interactions/contexts, while those with larger base kernel sizes emphasize longer-term/global contexts. The output of these Basic Units, denoted as variable b(s), essentially contains contextual information in different temporal ranges. The final variable in Figure 3(a) is denoted as h – the aggregated representation constructed by fusing information from all b(s). This latent embedding effectively utilizes the most representative knowledge contained in different contextual scales represented by b(s), thereby best capturing the unique characteristics of input samples.
> >
> > We have considered this suggestion and made better clarifications about this Figure, as well as provide more insights in Section 2.2 of our manuscript.
> >
> > In conclusion, we hope our responses have effectively addressed all your inquiries. We appreciate your positive evaluations of our paper. Upon the resolution of any remaining concerns, we earnestly hope for your consideration in adjusting your score and providing support. Thank you once more for your time and thoughtful consideration.
> >
> >
> > [1] Jean-Yves Franceschi, Aymeric Dieuleveut, and Martin Jaggi. Unsupervised scalable representation learning for multivariate time series. Advances in neural information processing systems, 32, 2019

---

### Official Review · Reviewer_sMvT · 2023-11-06

**Soundness:** 3 good
**Presentation:** 3 good
**Contribution:** 3 good
**Rating:** 6
**Confidence:** 4

**Summary:**

This paper proposes a new approach to time series analysis called CoInception, which integrates a noise-resilient sampling strategy with an efficient encoder architecture. The proposed method outperforms state-of-the-art methods in forecasting, classification, and abnormality detection tasks.

**Strengths:**

1. The proposed method outperforms the baselines in forecasting, classification, and abnormality detection tasks. This demonstrates the effectiveness of the proposed approach.

2.  The paper conducts comprehensive experiments to evaluate the efficacy of CoInception and analyze its behavior. This provides a thorough understanding of the proposed method and its strengths and weaknesses of each components.

3. The paper introduces a new approach to time series analysis that integrates a noise-resilient sampling strategy with an efficient encoder architecture. This approach has not been explored before.

**Weaknesses:**

1. The paper may not compare the proposed method with other state-of-the-art methods, making it difficult to assess its effectiveness. PatchTST, DLinear, TimesNet, all these SOTA methods are recommended to be included in the forecasting task.

2. The method section lacks originality, as it comprises three components from existing methods, and it lacks a coherent rationale for the integration of these three modules.

**Questions:**

1. Add PatchTST and DLinear baselines
2. The design of CoInception is not based on downstream tasks, why it works well on all three tasks? any insights?
3. What is the reason for using Inception as the backbone model?

---

> ### Author Response · Authors · 2023-11-14
> **Response to Reviewer sMvT**
>
> Thank you so much for your time and your evaluations of our paper. We are glad that you think our empirical studies are comprehensive and provide thorough understanding. Besides the responses, we have uploaded our revised manuscript and appendix, with all the revisions and additional details made based on your constructive feedback. We hope to discuss about the originality and novelty of our work which we try to clarify in the following responses to specific questions and concerns:
>
> *1. Regarding comparison with other state-of-the-art methods*
>
> We express our gratitude to you for sharing the three interesting studies, and we have completed the reading of these remarkably interesting works. We cite these references in our revised manuscript in the literature reviews section. We are working to add these methods into our experiments. However, due to the short rebuttal time and extensive experiments, we can not guarantee that such comparisons appear within the time period. Nevertheless, we will do our best to add the references to the experiments in the next version of our manuscript.
>
> *2. Regarding our CoInception backbone model and originality*
>
> Our sampling strategy is intentionally decoupled from the encoder architecture, and there are several benefits of making them modular.
> - For the sampling strategy, as you mentioned, we use an existing method - DWT low pass filter, hence it is not our main focus in this work. With the independent nature of this module, a different noise-canceling methodology, which is more suitable for particular tasks or datasets, can be installed in a plug-and-play manner, without affecting the function of the Encoder.
> - For the Encoder architecture, with this modular design, it carries no assumption about the data as well as the task being dealt. This nature, together with its design in accordance with our intention of balancing out between effectiveness and efficiency, explain for its robustness in both three tasks and various datasets discussed in our Experiment section. In addition, owing to the modular nature, any future studies can refer to our work and be benefited from our Encoder architecture without tied assumptions.
>
> That being said, inside each of these modules, there are strict connections between sub-components.
> - Regarding the sampling strategy, it should not be viewed in isolation within our framework. Instead, it should be integrated with our objectives, as represented by the loss functions. In addition to temporal-wise and instance-wise loss functions, which align samples from raw and disturbed series, we introduce an additional term acting as a regularization factor. This serves to stabilize the training process and mitigate the over-smoothing effect. The combination of the sampling stage and loss functions guides the CoInception framework in learning noise-resilient representations.
>
> - Concerning the Inception and Dilated Convolution architecture, their interaction ensures enhanced efficiency and effectiveness. The stacked Dilated Convolutional architecture has a notable weakness related to the selection of dilation factors. When these factors are too small, the parameter-efficient gain from dilation diminishes, and if they are too large, the framework overly focuses on a broad range of contextual information, potentially neglecting local details due to skipped spatial locations and limitations on the number of kernel filters used.
> To address this challenge, we incorporate the Inception idea into our Encoder. The Inception design naturally serves as a solution to automate the incorporation of various dilation factors in a single layer. By introducing different dilation factors of varying sizes into a single layer, our aim is to capture both local and global context without going too deep vertically into the number of network layers, while also reaching a large receptive field.
>
> In conclusion, we hope that our responses have sufficiently addressed all of your questions. We appreciate your favorable evaluations of our paper. Once all your concerns have been resolved, we sincerely hope for your consideration in increasing your score and offering your support. Thank you again for your time and consideration.

---

### Author Response · Authors · 2023-11-14
**General response**

We thank the AC and all reviewers for dedicating time and effort to review our paper.

**To all Reviewers**: Below we first outline our contributions with this study to make our paper clearer. Subsequently, we address here any common questions or confusion surrounding our work. Following this, we make our individual responses to each reviewer address the reviewers' concerns. Besides the responses, we have uploaded our revised manuscript and appendix, with all the revisions and additional details made based on Reviewers’ constructive feedback. We are happy to discuss with you further if you have any other questions.

Regarding the contributions of this study, we agree that the DWT Low Pass filter and the Inception, Dilated Convolution architectures have been subjects of extensive studies. Nevertheless, we emphasize that our novelty lies not in these individual techniques but rather in the unique motivations and methodologies we employ to integrate and utilize these established methods. Our major contributions stem from the novel ways we combine these techniques, offering a fresh perspective and approach to their application.

To provide specificity, we summarize our key contributions as follows:
- For the DWT Low Pass Filter, we leverage this technique as a part of our training strategy to generate noise-resilient representations for time series data. Our motivation is rooted in disregarding irrelevant high-frequency noise from signals of interest, leading the framework to focus on the genuine characteristics of the underlying series efficiently. The DWT-based low-pass filter presents a suitable solution, offering a smoother, cleaner version of raw input series without introducing additional training parameters or computational overheads. The sampling strategy, exclusively employed during training, is seamlessly integrated with our objectives represented by loss functions.
- Beside temporal-wised and instance-wised loss functions to align samples coming from raw series and disturbed ones, we additionally propose a new Triplet-based term serving as regularization factor. While striving to preserve both noise-resilience and contextual-invariance characteristics of learnt representations, we recognise a potential breakdown in the learning process, which is the over-smoothing effect in the circumstance low-pass filter eliminates too much high-frequency information. In such a situation, the regularization term helps our framework automatically soften the requirement of alignment between raw and perturbed series’ representations.
- Regarding the Inception and Dilated Convolution architecture, our motivation is to strike a balance between efficiency and effectiveness. The stacked Dilated Convolutional architecture achieves efficiency through lower-scale network parameters while ensuring robustness with a large accumulative receptive field. However, the challenge lies in selecting dilation factors, as too small or too large factors compromise parameter efficiency or focus excessively on long-range contextual information, respectively. The Inception design naturally addresses this by automating the incorporation of different dilation factors in a single layer, aiming to capture both local and global context without unnecessary depth in the network layers.


This response along with our individual replies to each reviewer thoroughly addresses their concerns. Given the importance of our two individual contributions in noise-resiliency training strategy and CoInception architecture, along with their joint effectiveness when combined together in various tasks, we sincerely hope the reviewers consider our responses and support our work.
We are more than welcome to provide further clarifications, if they are needed to emphasize the importance of our paper. We appreciate the AC's time and effort in managing the review discussion, and we value the feedback and comments from all reviewers. We look forward to additional responses and the chance to clarify aspects of our work for further discussion.